# FEATURE-SPECIFIC COEFFICIENTS OF DETERMINATION IN TREE ENSEMBLES

## ABSTRACT

Tree ensemble methods provide promising predictions with models difficult to interpret. Recent introduction of Shapley values for individualized feature contributions, accompanied with several fast computing algorithms for predicted values, shows intriguing results. However, individualizing coefficients of determination, aka $R^2$, for each feature is challenged by the underlying quadratic losses, although these coefficients allow us to comparatively assess single feature's contribution to tree ensembles. Here we propose an efficient algorithm, Q-SHAP, that reduces the computational complexity to polynomial time when calculating Shapley values related to quadratic losses. Our extensive simulation studies demonstrate that this approach not only enhances computational efficiency but also improves the estimation accuracy of feature-specific coefficients of determination.

## 1 INTRODUCTION

Models built with tree ensembles are powerful but often complicated, making it challenging to understand the influence of inputs. Feature importance plays a critical role in demystifying these models and enhancing their interpretability by assigning each input feature a score. This is crucial in domains like healthcare and biomedicine, where trust and interpretation of the model are essential (Stiglic et al., 2020; Bussmann et al., 2021). Common feature importance measures like gain can be inconsistent (Lundberg and Lee, 2017b) while permutation importance lacks theoretical foundations (Ishwaran, 2007).

Shapley values, derived from cooperative game theory and introduced by Shapley (1953), offer a robust method for the fair distribution of payoffs generated by a coalition of players. This can be analogously applied to assess the contribution of each feature in a machine learning model. It ensures that each feature's contribution is assessed by considering all possible combinations of features, thereby providing a comprehensive understanding of feature impacts. Recent applications of Shapley values have focused on local interpretation (Lundberg and Lee, 2017a;b; Chau et al., 2022), where they are employed to examine the influence of individual features on specific predictions. Nonetheless, there are numerous scenarios where global importance is preferred, such as analysis of the role of a feature across the entire dataset (Molnar, 2020; Covert et al., 2020).

Among the works that compute Shapley values in a global context, a popular approach is to use model variance decomposition. Lipovetsky and Conklin (2001) decomposed $R^2$ in linear regression, offering consistent interpretations even in the presence of multicollinearity. Owen and Prieur (2017) also conducted a conceptual analysis of Shapley values for the variance. However, computation remains a significant challenge, as the calculation of Shapley values grows exponentially with the number of features. To address this issue, several Monte Carlo-based methods have been proposed to effectively reduce the computational burden (Song et al., 2016; Covert et al., 2020; Williamson and Feng, 2020).

Although Monte Carlo-based, model-agnostic methods are more efficient than brute-force approaches, they are still computationally intensive, especially when dealing with high-dimensional data that requires extensive feature permutation sampling to ensure consistency (Lundberg and Lee, 2017a; Lundberg et al., 2020). This challenge has prompted the development of methods that leverage the specific structures of tree-based models. However, much of the focus has been on explaining individual predictions, as seen with TreeSHAP (Lundberg and Lee, 2017b), FastTreeSHAP (Yang,

2021), and LinearTreeSHAP (Bifet et al., 2022). Bénard et al. (2022) considered population-level importance using $R^2$, specifically tailored for random forests (Breiman, 2001).

To the best of our knowledge, there is no available method to calculate Shapley values of quadratic losses by leveraging structures of decision trees for fast computation. In this paper, we propose Q-SHAP, which can decompose quadratic terms of predicted values of a decision tree into each feature's attribute in polynomial time. It leads to fast computation of feature-specific $R^2$ for a decision tree. We also extend our approach to Gradient Boosted Decision Trees.

The rest of the paper is structured as follows. In Section 2, we provide a brief overview of Shapley values of $R^2$. In Section 3, we present our proposed algorithm Q-SHAP to calculate Shapley values of $R^2$ in polynomial time for single trees, and then extend the approach for tree ensembles in Section 4. We justify the efficacy and efficiency of the algorithm using extensive simulations in Section 5 and real data analysis in high dimension in Section 6. We conclude the paper with a discussion in Section 7.

## 2  Shapley Values of $R^2$ for Individual Features

### 2.1  Model specification

Here we investigate a specific label $Y$ and its explainability by a full set of $p$ features $X = (X_1, X_2, \cdots, X_p)$. For any subset $F \subseteq \mathcal{P} = \{1, 2, \cdots, p\}$, we define the corresponding set of features as $X_F = (X_j)_{j \in F}$.

Suppose that, for any set of features $X_F$, an oracle model $m_F$ can be built such that, for any specific value $x = (x_1, x_2, \cdots, x_p)$,

$$m_F(x) = E[Y|X_F = (x_j)_{j \in F}].$$

The Shapley value of $j$-th feature, in terms of its contribution to the total variation, is defined as

$$\phi_{\rho^2, j} = \frac{1}{p \, var(m_\emptyset)} \sum_{F \subseteq \mathcal{P} \setminus \{j\}} \left( var(m_{F \cup \{j\}}) - var(m_F) \right) \Big/ \binom{p-1}{|F|}, \tag{1}$$

where $|F|$ is the number of features in $F$. The term $var(m_{F \cup \{j\}})$ is the variance explained by feature set $F \cup \{j\}$ and the term $var(m_F)$ is the variance explained solely by set $F$. This definition is analogous to Covert et al. (2020) and Williamson and Feng (2020). By averaging over all possible feature combinations, the Shapley values are the only solution that satisfies the desired properties of symmetry, efficiency, additivity, and dummy (Shapley, 1953).

### 2.2  Empirical estimation

Suppose we have a set of data with sample size $n$ observed for both label and features as

$$\mathbf{Y} = (y_1, y_2, \cdots, y_n),$$
$$\mathbf{X} = (\mathbf{X}_{\cdot 1}, \mathbf{X}_{\cdot 2}, \cdots, \mathbf{X}_{\cdot p}) = \left( \mathbf{x}_{1\cdot}^T, \mathbf{x}_{2\cdot}^T, \cdots, \mathbf{x}_{n\cdot}^T \right)^T.$$

Accordingly, we denote the observed data of features in subset $F$ as

$$\mathbf{X}_{\cdot F} = (\mathbf{X}_{\cdot j})_{j \in F}.$$

Suppose that, for each subset $F$ of features, a single optimal model $\hat{m}_F$ is built on data $(\mathbf{Y}, \mathbf{X}_{\cdot F})$. Then the $i$-th label can be predicted with

$$\hat{y}_i(\mathbf{X}_{\cdot F}) = \hat{m}_F(\mathbf{x}_{i\cdot}).$$

### 2.3  From $R^2$ to a Quadratic Loss

We will establish the connection of $R^2$ to a quadratic loss through equation (1). We define the quadratic loss on the optimal model $\hat{m}_F$ as

$$Q_F = \sum_{i=1}^n (y_i - \hat{m}_F(\mathbf{x}_{i\cdot}))^2 \tag{2}$$

for any set of features $F$. With $m_\emptyset(\mathbf{x}_{i\cdot}) = \bar{y}$, we have $Q_\emptyset = \sum_{i=1}^{n}(y_i - \bar{y})^2$. Following the law of total variance, we can estimate $var(m_F)$ by

$$\widehat{var}(m_F) = (Q_\emptyset - Q_F)\big/n.$$

Thus, an empirical estimate of (1) is

$$\phi_{R^2,j} = -\frac{1}{pQ_\emptyset} \sum_{F \subseteq \mathcal{P}\setminus\{j\}} (Q_{F\cup\{j\}} - Q_F) \bigg/ \binom{p-1}{|F|},$$

which is proportional to a Shapley value for the sum of squared errors, i.e., the quadratic loss in (2).

## 2.4 FROM QUADRATIC LOSS TO Q-SHAP

We now further reduce Shapley values of the sum of squared errors to Shapley values of linear and quadratic terms of predicted values. Expanding the loss function in (2), we can rewrite,

$$\phi_{R^2,j} = -\frac{1}{pQ_\emptyset} \sum_{F \subseteq \mathcal{P}\setminus\{j\}} \sum_{i=1}^{n} \big(\hat{m}^2_{F\cup j}(\mathbf{x}_{i\cdot}) - \hat{m}^2_F(\mathbf{x}_{i\cdot}) - 2(\hat{m}_{F\cup j}(\mathbf{x}_{i\cdot}) - \hat{m}_F(\mathbf{x}_{i\cdot}))y_i\big) \bigg/ \binom{p-1}{|F|}.$$

To calculate this, we define the Shapley value for each sample $i$ as,

$$
\begin{aligned}
\phi_{R^2,j}(\mathbf{x}_{i\cdot}) \;=\; & -\frac{1}{pQ_\emptyset} \sum_{F \subseteq \mathcal{P}\setminus\{j\}} \big(\hat{m}^2_{F\cup j}(\mathbf{x}_{i\cdot}) - \hat{m}^2_F(\mathbf{x}_{i\cdot})\big) \bigg/ \binom{p-1}{|F|} \\
& +\frac{2y_i}{pQ_\emptyset} \sum_{F \subseteq \mathcal{P}\setminus\{j\}} \big(\hat{m}_{F\cup j}(\mathbf{x}_{i\cdot}) - \hat{m}_F(\mathbf{x}_{i\cdot})\big) \bigg/ \binom{p-1}{|F|},
\end{aligned}
$$

which is a linear combination of two sets of Shapley values, i.e., Shapley values of predicted value $\hat{m}_F$, which are ready to be calculated (Lundberg and Lee, 2017b; Yang, 2021; Bifet et al., 2022), and Shapley values of the quadratic term of predicted value $\hat{m}^2_F$, i.e.,

$$\phi_{\hat{m}^2,j}(\mathbf{x}_{i\cdot}) \;=\; \frac{1}{p} \sum_{F \subseteq \mathcal{P}\setminus\{j\}} \big(\hat{m}^2_{F\cup j}(\mathbf{x}_{i\cdot}) - \hat{m}^2_F(\mathbf{x}_{i\cdot})\big) \bigg/ \binom{p-1}{|F|}, \tag{3}$$

for which we will develop the algorithm Q-SHAP to calculate. For the rest of the paper, we will focus on computing the Shapley values in Equation (3) in polynomial time for tree-based models and carrying it over to calculate feature-specific $R^2$.

# 3 THE ALGORITHM Q-SHAP FOR SINGLE TREES

Unlike most regression problems that can yield infinitely many predictions across the diverse input space $\mathbf{X}$, decision trees restrict predictions to a finite set, specifically to the values at each leaf node. This nature revitalizes hope in previously unattainable solutions to Shapley values in decision tree-based models. The main idea of our algorithm lies in the fact that Shapley values with various targets such as predictions and various loss functions, are essentially weighted functions of the leaf predictions. While Lundberg et al. (2020) suggests that explaining the loss function for a "path-dependent" algorithm is challenging, we provide an exact solution to decomposing quadratic losses using our Q-SHAP. Q-SHAP works for tree ensembles. For simplicity, we here illustrate it with a single decision tree.

## 3.1 NOTATIONS

We assume the underlying decision tree has the maximum depth at $D$ and a total of $L$ leaves, and use $l$ to denote a specific leaf. We further introduce a dot product for polynomials for subsequent calculation. For two polynomials $A(z) = \sum_{i=0}^{n} a_i z^i$ and $B(z) = \sum_{i=0}^{n} b_i z^i$, we define their dot product as $A(z) \cdot B(z) = \sum_{i=0}^{n} a_i b_i$.

## 3.2 Distributing the Prediction to Leaves

Decision trees match each data point to one leaf for prediction. However, for our prediction defined on any subset $F$, a data point $\mathbf{x}_{i\cdot}$ can fall into multiple leaves due to the uncertainty by unspecified features $\mathcal{P} \backslash F$. We can calculate $\hat{m}_F(\mathbf{x}_{i\cdot})$, following TreeSHAP, as the empirical mean by aggregating the weighted prediction on each leaf,

$$\hat{m}_F(\mathbf{x}_{i\cdot}) = \sum_l \hat{m}_F^l(\mathbf{x}_{i\cdot}), \tag{4}$$

where $\hat{m}_F^l$ is the weighted prediction from leaf $l$ in a tree built on feature set $F$.

Given an oracle tree built on all available features, we try to recover the oracle tree for a subset of features without rebuilding, following Bifet et al. (2022) and Karczmarz et al. (2022). For example, let us take the tree built on two features $X_1$ and $X_2$ as shown in Figure 1. Figure 2 can be viewed as the oracle tree built solely on feature $X_1$, and hence $X_2$ no longer exists in the tree. Therefore, we replace it with a pseudo internal node to preserve the structure of the original full oracle tree and pave the way for further formulation.

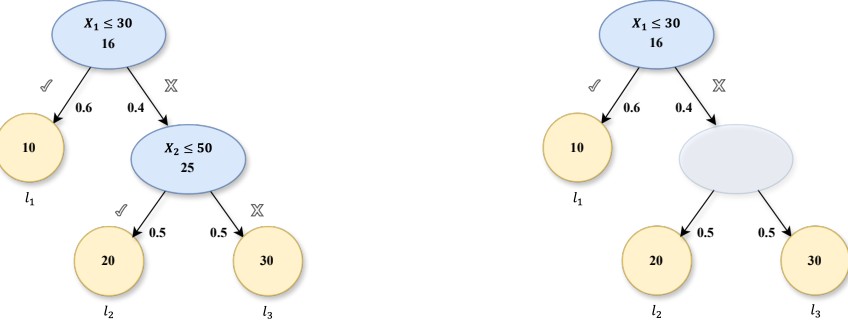

Figure 1: Decision tree built on $X_1$ and $X_2$      Figure 2: Hypothetical decision tree with $X_1$ only

With a data point $\mathbf{x}_{i\cdot} = (40, 25)$, we illustrate the calculation in Equation (4) by first calculating the predicted value for the tree in Figure 2, $\hat{m}_{\{1\}}(\mathbf{x}_{i\cdot}) = \hat{m}_{\{1\}}^{l_1}(\mathbf{x}_{i\cdot}) + \hat{m}_{\{1\}}^{l_2}(\mathbf{x}_{i\cdot}) + \hat{m}_{\{1\}}^{l_3}(\mathbf{x}_{i\cdot}) = 0 \times 10 + 0.5 \times 20 + 0.5 \times 30$. When the tree is built with an additional feature $j = 2$ as shown in Figure 1, we have the predicted value $\hat{m}_{\{1,2\}}(\mathbf{x}_{i\cdot}) = \hat{m}_{\{1,2\}}^{l_1}(\mathbf{x}_{i\cdot}) + \hat{m}_{\{1,2\}}^{l_2}(\mathbf{x}_{i\cdot}) + \hat{m}_{\{1,2\}}^{l_3}(\mathbf{x}_{i\cdot}) = \mathbf{1} \times 0 \times 10 + \mathbf{0.5^{-1}} \times 0.5 \times 20 + \mathbf{0} \times 0.5 \times 30$ where the bold numbers reweight $\hat{m}_{\{1\}}^l(\mathbf{x}_{i\cdot})$ for $\hat{m}_{\{1,2\}}^l(\mathbf{x}_{i\cdot})$. Let us take a closer look at these weights, each corresponding to one leaf. For leaf $l_1$, the weight is 1 since the newly added feature $X_2$ is not involved in its path and the reweighted prediction remains as zero. For leaf $l_2$, the reweighted prediction is lifted up with the weight inversely proportional to the previous probability because $\mathbf{x}_{i\cdot}$ follows its path to the leaf with probability 1. On the other hand, although the path to leaf $l_3$ includes the newly added feature but $\mathbf{x}_{i\cdot}$ doesn't follow this path, resulting in a weight at 0. Next we will generalize such a reweighting strategy to calculate (4) for trees with different sets of features.

Denote $F^l$ the features involved in the path to leaf $l$ and $F^l(\mathbf{x}_{i\cdot})$ the subset of $F^l$ whose decision criteria are satisfied by $\mathbf{x}_{i\cdot}$. Note that each feature $j \in F^l$ may appear multiple times in the path to leaf $l$ so we denote $n_{j,c}^l$ the number of samples passing through the node which is attached to the $c$-th appearance. We similarly define $n_{j,c}^l(\mathbf{x}_{i\cdot})$ for each feature $j \in F^l(\mathbf{x}_{i\cdot})$.

For any feature $j \in \mathcal{P}$, we can define the weight function based on a partition of $\mathcal{P}$ into three subsets $F^l(\mathbf{x}_{i\cdot})$, $F^l \backslash F^l(\mathbf{x}_{i\cdot})$, and $\mathcal{P} \backslash F^l$,

$$w_j^l(\mathbf{x}_{i\cdot}) \triangleq \begin{cases} \prod_c \frac{n_{j,c}^l}{n_{j,c}^l(\mathbf{x}_{i\cdot})}, & \text{if } j \in F^l(\mathbf{x}_{i\cdot}); \\ 0, & \text{if } j \in F^l \backslash F^l(\mathbf{x}_{i\cdot}); \\ 1, & \text{if } j \in \mathcal{P} \backslash F^l. \end{cases}$$

Therefore, for $j \notin F$, we have

$$\hat{m}_{F \cup j}^l(\mathbf{x}_{i\cdot}) = w_j^l(\mathbf{x}_{i\cdot}) \hat{m}_F^l(\mathbf{x}_{i\cdot}).$$

Recursive application of the above formula leads to

$$\hat{m}_F^l(\mathbf{x}_{i\cdot}) = \prod_{k \in F} w_k^l(\mathbf{x}_{i\cdot})\hat{m}_\emptyset^l,$$

where $\hat{m}_\emptyset^l = \hat{m}^l \frac{n^l}{n}$ with $n^l$ the sample size at leaf $l$, $n$ the total sample size, and $\hat{m}^l$ the predicted value at leaf $l$ based on the model built on all features.

When $F = \emptyset$, the above result reduces to $\hat{m}_\emptyset(\mathbf{x}_{i\cdot}) = \sum_l \hat{m}^l \frac{n^l}{n}$, so the optimal prediction is just the mean for all data points, which is consistent with $\hat{m}_\emptyset(\mathbf{x}_{i\cdot}) = \bar{y}$.

We can rewrite (3) as

$$\phi_{\hat{m}^2, j}(\mathbf{x}_{i\cdot})$$

$$= \frac{1}{p} \sum_{F \subseteq \mathcal{P} \backslash \{j\}} \left( \sum_l \left( w_j^{l2}(\mathbf{x}_{i\cdot}) - 1 \right) \hat{m}_\emptyset^{l2} \prod_{k \in F} w_k^{l2}(\mathbf{x}_{i\cdot}) \right) \Big/ \binom{p-1}{|F|}$$

$$+ \frac{2}{p} \sum_{F \subseteq \mathcal{P} \backslash \{j\}} \left( \sum_{l_1 \neq l_2} \left( w_j^{l_1}(\mathbf{x}_{i\cdot}) w_j^{l_2}(\mathbf{x}_{i\cdot}) - 1 \right) \hat{m}_\emptyset^{l_1} \hat{m}_\emptyset^{l_2} \prod_{k \in F} w_k^{l_1}(\mathbf{x}_{i\cdot}) w_k^{l_2}(\mathbf{x}_{i\cdot}) \right) \Big/ \binom{p-1}{|F|}$$

$$\triangleq T_{1,j}(\mathbf{x}_{i\cdot}) + 2T_{2,j}(\mathbf{x}_{i\cdot}).$$

We further define, for leaves $l_1$ and $l_2$,

$$T_j^{l_1 l_2}(\mathbf{x}_{i\cdot})$$

$$= \frac{1}{p} \sum_{F \subseteq \mathcal{P} \backslash \{j\}} \left( \left( w_j^{l_1}(\mathbf{x}_{i\cdot}) w_j^{l_2}(\mathbf{x}_{i\cdot}) - 1 \right) \hat{m}_\emptyset^{l_1} \hat{m}_\emptyset^{l_2} \prod_{k \in F} w_k^{l_1}(\mathbf{x}_{i\cdot}) w_k^{l_2}(\mathbf{x}_{i\cdot}) \right) \Big/ \binom{p-1}{|F|}, \quad (5)$$

and we have $T_{2,j}(\mathbf{x}_{i\cdot}) = \sum_{l_1 \neq l_2} T_j^{l_1 l_2}(\mathbf{x}_{i\cdot})$, $T_{1,j}(\mathbf{x}_{i\cdot}) = \sum_l T_j^{ll}(\mathbf{x}_{i\cdot})$. Therefore, we will focus on the calculation of $T_j^{l_1 l_2}(\mathbf{x}_{i\cdot})$ in (5) the rest of this section.

We can reduce the calculation of $\prod_{k \in F} w_k^{l_1}(\mathbf{x}_{i\cdot}) w_k^{l_2}(\mathbf{x}_{i\cdot})$ in (5) by only calculating $\prod_{k \in F_-} w_k^{l_1}(\mathbf{x}_{i\cdot}) w_k^{l_2}(\mathbf{x}_{i\cdot})$ with $F_- = F \cap (F^{l_1} \cup F^{l_2})$, because $\prod_{k \in F \backslash F_-} w_k^{l_1}(\mathbf{x}_{i\cdot}) w_k^{l_2}(\mathbf{x}_{i\cdot}) = 1$. In combination with the proposition below, computation in (5) can be dramatically reduced from the full feature set $\mathcal{P}$ to a set only related to the corresponding leaves in a tree.

**Proposition 1** *For any well-defined $p, n, |F|$,*

$$\sum_{k=0}^{p-n} \frac{\binom{p-n}{k}}{p \binom{p-1}{|F|+k}} = \frac{1}{n \binom{n-1}{|F|}}.$$

We leave the proof of Proposition 1 in Appendix A. Further denote $n_{12} = |F^{l_1} \cup F^{l_2}|$ and a polynomial of $z$, $P^{l_1 l_2}(z) = \prod_{k \in F^{l_1} \cup F^{l_2} \backslash j} (z + w_k^{l_1}(\mathbf{x}_{i\cdot}) w_k^{l_2}(\mathbf{x}_{i\cdot}))$. We then define a coefficient polynomial $C_{n_{12}}(z) = \frac{1}{\binom{n_{12}-1}{0}} z^0 + \frac{1}{\binom{n_{12}-1}{1}} z^1 + \cdots + \frac{1}{\binom{n_{12}-1}{n_{12}-1}} z^{n_{12}-1}$.

**Theorem 1**

$$T_j^{l_1 l_2}(\mathbf{x}_{i\cdot}) = \frac{1}{n_{12}} (w_j^{l_1}(\mathbf{x}_{i\cdot}) w_j^{l_2}(\mathbf{x}_{i\cdot}) - 1) \hat{m}_\emptyset^{l_1} \hat{m}_\emptyset^{l_2} [C_{n_{12}}(z) \cdot P^{l_1 l_2}(z)].$$

**Proof**. With Proposition 1, we can write (5) as

$$T_j^{l_1 l_2}(\mathbf{x}_{i\cdot}) = \frac{1}{n_{12}} (w_j^{l_1}(\mathbf{x}_{i\cdot}) w_j^{l_2}(\mathbf{x}_{i\cdot}) - 1) \hat{m}_\emptyset^{l_1} \hat{m}_\emptyset^{l_2} \sum_{t=0}^{n_{12}-1} \frac{1}{\binom{n_{12}-1}{t}} \sum_{F \subseteq F^{l_1} \cup F^{l_2} \backslash j}^{|F|=t} \prod_{k \in F} w_k^{l_1}(\mathbf{x}_{i\cdot}) w_k^{l_2}(\mathbf{x}_{i\cdot}).$$

We further notice that $\sum_{F \subseteq F^{l_1} \cup F^{l_2} \backslash j}^{|F|=t} \prod_{k \in F} w_k^{l_1}(\mathbf{x}_{i\cdot}) w_k^{l_2}(\mathbf{x}_{i\cdot})$ is the coefficient of $z^t$ in polynomial $P^{l_1 l_2}(z)$, hence the equation holds with $C_{n_{12}}(z)$ adjusting the weight based on the size of set $F$. ∎

We only need to consider feature $j \in |F^{l_1} \cup F^{l_2}|$ as, otherwise, we have $T_j^{l_1 l_2}(\mathbf{x}_{i\cdot}) = 0$ following the definition of $w_j^l(\mathbf{x}_{i\cdot})$. Note that, when there is a feature in set $F$ that doesn't belong to $F^{l_1}(\mathbf{x}_{i\cdot}) \cap F^{l_2}(\mathbf{x}_{i\cdot}) \backslash j$, we have $\prod_{k \in F} w_k^{l_1}(\mathbf{x}_{i\cdot}) w_k^{l_2}(\mathbf{x}_{i\cdot}) = 0$. Thus we can further simplify the term to

$$T_j^{l_1 l_2}(\mathbf{x}_{i\cdot})$$

$$= \frac{1}{n_{12}}(w_j^{l_1}(\mathbf{x}_{i\cdot}) w_j^{l_2}(\mathbf{x}_{i\cdot}) - 1)\hat{m}_\emptyset^{l_1}\hat{m}_\emptyset^{l_2} \sum_{t=0}^{n_{12}-1} \frac{1}{\binom{n_{12}-1}{t}} \sum_{F \subseteq F^{l_1}(\mathbf{x}_{i\cdot}) \cap F^{l_2}(\mathbf{x}_{i\cdot}) \backslash j}^{|F|=t} \prod_{k \in F} w_k^{l_1}(\mathbf{x}_{i\cdot}) w_k^{l_2}(\mathbf{x}_{i\cdot}).$$

Consequently, the evaluation of $P^{l_1 l_2}(z)$ can be reduced to a much smaller set.

### 3.3 THE ALGORITHM

In this section, we will introduce our algorithms. Theorem 1 demonstrates that we can construct a polynomial form of the NP-problem. Now we introduce a fast and stable evaluation for the dot product of a coefficient polynomial $C(z)$ where we know the coefficients and a polynomial $P(z)$ with a known product form, involved in Theorem 1.

**Proposition 2** *Let $\omega$ be a vector of the complex $n$-th roots of unity whose element is $\exp(\frac{2k\pi i}{n})$ for $k = 0, 1., \ldots, n-1$, $c$ the coefficient vector of $C(z)$, and IFFT the Inverse Fast Fourier Transformation. Then*

$$C(z) \cdot P(z) = P(\omega)^T IFFT(c).$$

The proof of Proposition 2 is shown in Appendix A. We facilitate the computation via the complex roots of unity because of their numerical stability and fast operations in matrix multiplications. Due to the potential issue of ill condition, especially at large degrees, our calculation avoids inversion of the Vandermonde matrices, although it has been proposed to facilitate the computing by Bifet et al. (2022). In addition, for each sample size $n$, we only need to calculate IFFT$(c)$ once, up to order $D$ in $O(n \log(n))$ operations, and the results can be saved for the rest of calculation through Q-SHAP. Note that term $k$ and term $n-k$ in $P(w)$ are complex conjugates, and, for a real vector $c$, IFFT$(c)$ also has the conjugate property for paired term $k$ and term $n-k$. Consequently, the dot product of $P(\omega)$ and IFFT$(c)$ inherits the conjugate property and its imaginary parts are canceled upon addition. Therefore, we only need evaluate the dot product at half of the $n$ complex roots.

We can aggregate the values of leaf combinations to derive the Shapley values of squared predictions using Q-SHAP as in Algorithm 1 and then calculate the Shapley values of $R^2$ using RSQ-SHAP as in Algorithm 2. The calculation of feature-specific $R^2$ uses the iterative Algorithm 1 instead of a recursive one. As detailed in Appendix E, the time complexity of the algorithm is $O(L^2 D^2)$ for a single tree, which is super fast when the maximum tree depth is not too large.

---

**Algorithm 1 Q-SHAP**

**Q-SHAP**$(\mathbf{x}_{i\cdot})$
Initialize $T[j] = 0$ for $j = 1, \cdots, p$
**for** $l_1 \in$ index set $0, \ldots, L-1$ **do**
    **for** $l_2 \in$ index set $l_1, \ldots, L-1$ **do**
        Let $n_{12} = |F^{l_1} \cup F^{l_2}|$
        **for** $j \in F^{l_1} \cup F^{l_2}$ **do**
            Let $t[j] = \frac{1}{n_{12}}[w_j^{l_1}(\mathbf{x}_{i\cdot}) w_j^{l_2}(\mathbf{x}_{i\cdot}) - 1]\hat{m}_\emptyset^{l_1}\hat{m}_\emptyset^{l_2}[C_{n_{12}}(z) \cdot P^{l_1 l_2}(z)]$
            **if** $l_1 \neq l_2$ **then**
                $T[j] = T[j] + 2t[j]$
            **else**
                $T[j] = T[j] + t[j]$
            **end if**
        **end for**
    **end for**
**end for**
return $T = (T[1], T[2], \cdots, T[p])$

---

---
**Algorithm 2 RSQ-SHAP**

---

$\textbf{RSQ-SHAP}(j) = -\frac{1}{Q_\emptyset}\Sigma_{i=1}^n \left\{\textbf{Q-SHAP}(\mathbf{x}_{i\cdot})[j] - 2y_i\textbf{SHAP}(\mathbf{x}_{i\cdot})[j]\right\}$

---

## 4 THE ALGORITHM Q-SHAP FOR TREE ENSEMBLES FROM BOOSTING

Tree ensembles from Gradient Boosted Machines (GBM) (Friedman, 2001) greatly improve predictive performance by aggregating many weak learners (Chen and Guestrin, 2016; Ke et al., 2017; Prokhorenkova et al., 2018). Each tree, say tree $k$, is constructed on the residuals from the previous tree, i.e., tree $k-1$. We assume that there are a total of $K$ trees in the ensemble and the quadratic loss by the first $k$ trees, with all features in $\mathcal{P}$, is $Q_\mathcal{P}^{(k)}$. Denoting $Q_\mathcal{P}^{(0)} = Q_\emptyset$, the $k$-th tree reduces the loss by

$$\Delta Q_\mathcal{P}^{(k)} = Q_\mathcal{P}^{(k-1)} - Q_\mathcal{P}^{(k)},$$

with the tree ensemble reducing the total loss by

$$Q_\emptyset - Q_\mathcal{P}^{(K)} = \sum_{k=1}^K \Delta Q_\mathcal{P}^{(k)}.$$

Per our interest in feature-specific $R^2$, we resort to the quadratic loss defined as the sum of squared errors in (2).

On the other hand, the $k$-th tree provides the prediction $\hat{m}_\mathcal{P}^{(k)}(\mathbf{x}_{i\cdot})$. Therefore, the prediction by the first $k$ trees can be recursively calculated as $\hat{y}_i^{(k)}(\mathbf{X}) = \hat{y}_i^{(k-1)}(\mathbf{X}) + \alpha\hat{m}_\mathcal{P}^{(k)}(\mathbf{x}_{i\cdot})$, where $\alpha$ is the learning rate and $\hat{y}_i^{(0)}(\mathbf{X}) \equiv \bar{y}$. Note that the residuals after building $(k-1)$ tree are $\{r_i^{(k-1)} = y_i - \hat{y}_i^{(k-1)}(\mathbf{X}) : i = 1, 2, \cdots, n\}$, which are taken to build the $k$-th tree. Thus,

$$\Delta Q_\mathcal{P}^{(k)} = \sum_{i=1}^n (r_i^{(k-1)})^2 - \sum_{i=1}^n (r_i^{(k-1)} - \alpha\hat{m}_\mathcal{P}^{(k)}(\mathbf{x}_{i\cdot}))^2 = -\sum_{i=1}^n (\alpha^2\hat{m}_\mathcal{P}^{(k)2}(\mathbf{x}_{i\cdot}) - 2\alpha r_i^{(k-1)}\hat{m}_\mathcal{P}^{(k)}(\mathbf{x}_{i\cdot})).$$

Therefore, the decomposition of $\Delta Q_\mathcal{P}^{(k)}$ for feature-specific Shapley values in the $k$-th tree is again subject to the decomposition of two sets of values, i.e., both the predicted value $\hat{m}_\mathcal{P}^{(k)}(\mathbf{x}_{i\cdot})$ and its quadratic term, which can be similarly dealt with the algorithms proposed in the previous section.

## 5 SIMULATION STUDY

One of the challenges in assessing methods that explain predictions is the typical absence of a definitive ground truth. Therefore, to fairly demonstrate the fidelity of our methodology, we must rely on synthetic data that allows for the calculation of the theoretical Shapley values. Here we consider three different models,

  a.  $Y = 4X_1 - 5X_2 + 6X_3 + \epsilon$;
  b.  $Y = 4X_1 - 5X_2 + 6X_3 + 3X_1X_2 - X_1X_3 + \epsilon$;
  c.  $Y = 4X_1 - 5X_2 + 6X_3 + 3X_1X_2 - X_1X_2X_3 + \epsilon$.

All three features involved in the models are generated from Bernoulli distributions,

$$X_1 \sim Bernoulli(0.6), X_2 \sim Bernoulli(0.7), X_3 \sim Bernoulli(0.5).$$

We also simulate additional nuisance features independently from $Bernoulli(0.5)$ to make the total number of features $p = 100$ and $p = 500$, respectively. The error term $\epsilon$ is generated from $N(0, \sigma_\epsilon^2)$ with $\sigma_\epsilon$ at three different levels, i.e., 0.5, 1, and 1.5. The theoretical values of total $R^2$ and feature-specific $R^2$ are shown in Table 1 of Appendix B as well as indicated by dashed lines in Figure 3.

We evaluate the performance of three different methods, our proposed Q-SHAP, SAGE by Covert et al. (2020), and SPVIM by Williamson and Feng (2020), in calculating the feature-specific $R^2$ for

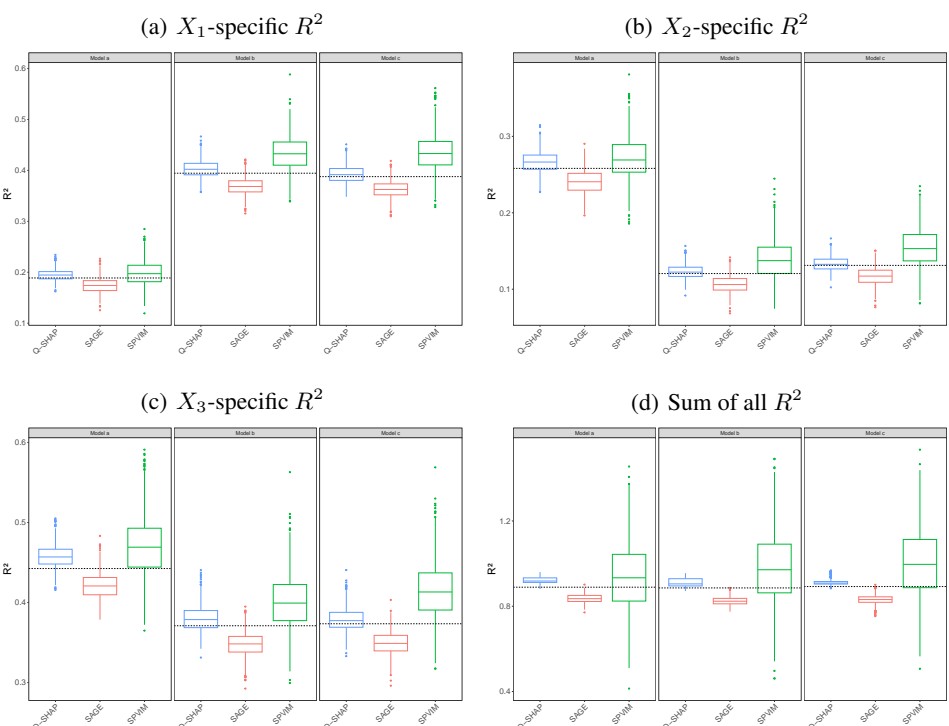

Figure 3: Boxplots of (a) $X_1$-specific, (b) $X_2$-specific, (c) $X_3$-specific, and (d) the sum of all feature-specific $R^2$ in the three models with $n = 1,000$, $p = 100$, and $\sigma_\epsilon = 1.5$. The dashed lines show the theoretical $R^2$.

the above three models with data sets of different sample sizes at $n = 500, 1000, 2000,$ and $5000$. We use package *sage-importance* for SAGE and package *vimpy* for SPVIM to calculate feature-specific Shapley values of total explained variance, which are divided by the total variance for corresponding feature-specific $R^2$ values.

For each setting, we generated 1,000 data sets. For each data set, we built a tree ensemble using XGBoost (Chen and Guestrin, 2016) with tuning parameters optimized via 5-fold cross-validation and grid search in a parameter space specified with the learning rate in $\{0.01, 0.05, 0.1\}$ and number of estimators in $\{50, 100, 200, 300, \cdots, 1000\}$. We fixed the maximum depth of models a, b, and c at 1, 2, and 3 respectively. Figure 3 show the calculated feature-specific $R^2$ for the first three features as well as the sum of all feature-specific $R^2$ for all three models with $n = 1000$, $p = 100$, and $\sigma_\epsilon = 1.5$. The results of the three models in other settings are shown in Appendix C. Overall, Q-SHAP provides a more stable and accurate calculation of feature-specific $R^2$ than the other two methods.

We divide all features into two groups, signal features (the first three) and nuisance features (the rest). For each group, we calculated the mean absolute error (MAE) by comparing feature-specific $R^2$ values to the theoretical ones in each of the 1,000 datasets and averaged MAE over the 1,000 datasets, shown in Figure 4. Note that, by limiting memory to 2GB, SAGE can only report $R^2$ for the data sets with sample size at 500 and 1,000.

For both signal and nuisance features, Q-SHAP and SAGE exhibit consistent behavior across all models. In contrast, SPVIM tends to bias the calculation, especially for small sample sizes, indicated by the rapid increase of MMAE when the sample size goes down. Among signal features, Q-SHAP has better accuracy than SAGE, followed by SPVIM in general. All methods tend to have better accuracy when sample size increases.

For the nuisance features, only SPVIM is biased away from 0. On the other hand, both Q-SHAP and SAGE have almost no bias for nuisance features across different sample sizes. For all three methods, $R^2$ of signal features tends to have a larger bias than nuisance features.

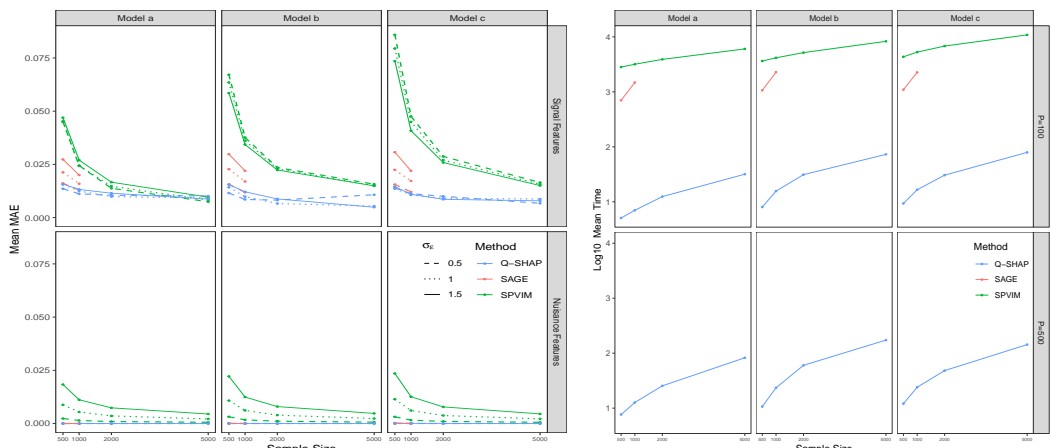

Figure 4: Mean MAE with $p = 100$    Figure 5: Running Time in $\log_{10}(\text{seconds})$

We compared the computational time of the three different methods by running all algorithms in parallel on a full node consisting of two AMD CPUs@2.2GHz with 128 cores and 256 GB memory. We unified the environment with the help of a Singularity container (Kurtzer et al., 2017) built under Python version 3.11.6. Due to the large size of the simulation, we limit all methods to a maximum wall time of 4 hours per dataset on a single core, with memory limited to 2 GB. The running times are shown in Figure 5. Both SAGE and SPVIM demanded a long time to compute even with only 100 features. Q-SHAP is hundreds of times faster than both SAGE and SPVIM in general and is the only method that can be completed when the dimension is 500 in constrained computation time and memory.

## 6 REAL DATA ANALYSIS

We illustrate the utility of Q-SHAP by applying it to predicting Gleason score, a grading prognosis of men with prostate cancer, with gene expressions. The dataset was obtained from The Cancer Genome Atlas Program (TCGA) (Weinstein et al., 2013), including 551 samples and 17,261 features. The Gleason score was retrieved through TCGAbiolinks (Colaprico et al., 2016) and further adjusted for age and race as potential confounding factors. The gene expression data was downloaded from UCSC Xena (Goldman et al., 2020) and preprocessed using SIGNET (Jiang et al., 2023).

We first constructed the tree ensemble using XGBoost with tuning parameters optimized via 5-fold cross-validation and random research in a parameter space specified with the number of trees in $\{50, 100, 500, 1000, 1500, 2000, 2500, 3000\}$, maximum depth in $\{1, 2, \cdots, 6\}$, and learning rate in $\{0.01, 0.05, 0.1\}$. As both SAGE and SPVIM cannot manage this large number of features, we only applied Q-SHAP to the tree ensemble built on 17,261 features, which took 11 minutes to compute and reported the sum of all feature-specific $R^2$ at $99.98\%$ which is equivalent to the model $R^2$. The 15 highest feature-specific $R^2$ values are reported in Figure 6.

To allow the application of both SAGE and SPVIM, we selected the top 100 features based on the result of Q-SHAP, and rebuilt the tree ensemble with the selected 100 features using XGBoost. The rebuilt tree ensemble reported a total of $R^2$ at 0.93. Calculating the 100 feature-specific $R^2$ values took about 10 seconds of Q-SHAP, 16 minutes of SAGE, and 67 hours of SPVIM. As shown in Figures 7-9, SPVIM tends to overstate the feature-specific $R^2$ although all of its feature-specific $R^2$ only sums up to 0.72, much lower than 0.93 reported by Q-SHAP. On the other hand, SAGE tends to underestimate the feature-specific $R^2$ and all its feature-specific $R^2$ also sums up to only 0.73, but its top 15 features match well with those by Q-SHAP. We also applied the three methods to two additional datasets, and the results are summarized in Appendix F. In summary, the real data analysis shows consistent results with the simulation study, confirming that Q-SHAP is superior in both computational time and the accuracy of feature-specific $R^2$.

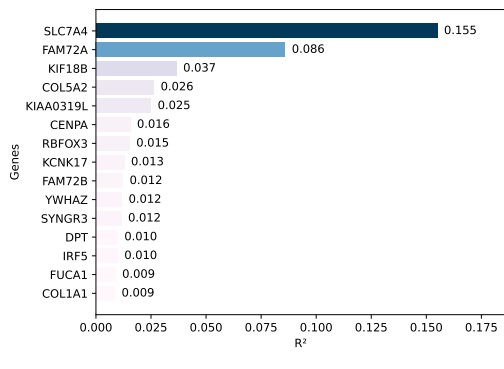

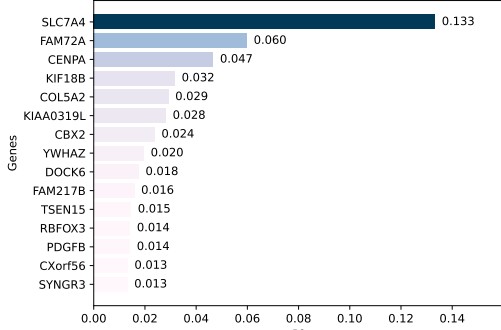

Figure 6: Top 15 of 17,261 $R^2$ by Q-SHAP

Figure 7: Top 15 of 100 $R^2$ by Q-SHAP

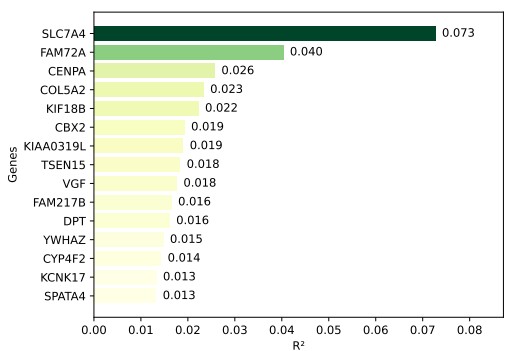

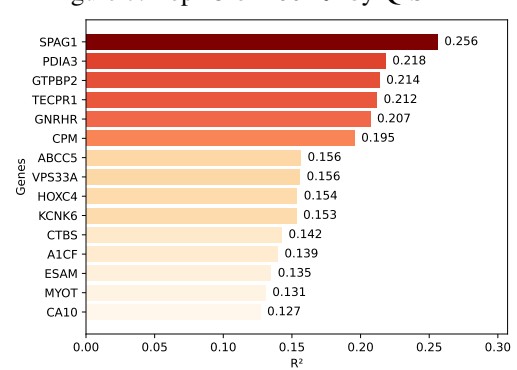

Figure 8: Top 15 of 100 $R^2$ by SAGE

Figure 9: Top 15 of 100 $R^2$ by SPVIM

# 7 CONCLUSION

The coefficient of determination, aka $R^2$, measures the proportion of the total variation explained by available features. Its additive decomposition, following Shapley (1953), provides an ideal evaluation of each feature's attribute to explain the total variation. However, the calculation of corresponding Shapley values is an NP-hard problem, and is further complicated by the complexities involved in building tree ensembles. Recently, several methods (Lundberg and Lee, 2017b; Yang, 2021; Bifet et al., 2022) have been developed to leverage the structure of tree-based models and provide computationally efficient algorithms to decompose the predicted values. However, decomposing $R^2$ demands the decomposition of a quadratic loss reduction by multiple trees. We have shown in Section 4 that we can attribute the total loss reduction by the tree ensemble to each single tree, and the tree-specific loss reductions are subject to further decomposition to each feature. However, decomposing a quadratic loss of a single tree needs work with the squared terms of predicted values, invalidating previously developed methods for predicted values. Thus we developed the Q-SHAP algorithm to calculate Shapley values of squared terms of predicted values in polynomial time. The algorithm works not only for $R^2$ but also for general quadratic losses. Ultimately, it may provide a framework for more general loss functions via approximation.

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

## A    PROOFS

We first establish the following lemma.

**Lemma 1**

$$\sum_{k=0}^{p-n} \binom{|F|+k}{k} \binom{p-1-|F|-k}{p-n-k} = \binom{p}{n}.$$

**Proof**. Using Gould's identity (Gould, 1972), we have

$$\sum_{k=0}^{p-n} \binom{|F|+k}{k} \binom{p-1-|F|-k}{p-n-k} = \sum_{k=0}^{p-n} \binom{k}{k} \binom{p-k-1}{p-n-k} = \sum_{k=0}^{p-n} \binom{p-k-1}{n-1} = \binom{p}{n},$$

where we used the Hockey-Stick Identity in the last step. ∎

**Proof of Proposition 1**. Through expansion and Lemma 1, we have

$$\begin{aligned}
\sum_{k=0}^{p-n} \frac{\binom{p-n}{k}\binom{n-1}{|F|}}{p\binom{p-1}{|F|+k}} &= \frac{1}{p} \sum_{k=0}^{p-n} \frac{(|F|+k)!(p-n)!(p-1-|F|-k)!(n-1)!}{(p-1)!(p-n-k)!k!|F|!(n-1-|F|!)} \\
&= \frac{1}{p} \frac{(p-n)!(n-1)!}{(p-1)!} \sum_{k=0}^{p-n} \binom{|F|+k}{k} \binom{p-1-|F|-k}{p-n-k} \\
&= \frac{1}{p} \frac{(p-n)!(n-1)!}{(p-1)!} \binom{p-1}{n-1} \frac{p}{n} = \frac{1}{n}.
\end{aligned}$$

∎

**Proof of Proposition 2**. We first rewrite the two polynomials

$$C(z) = V(z)c, \quad P(z) = V(z)a,$$

where $V(z)$ is the Vandermonde matrix for vector $z$, and $c$ and $a$ are the coefficients of polynomials $C(z)$ and $P(z)$, respectively. Then the inner product

$$C(z) \cdot P(z) = P(z) \cdot C(z) = a^T c = (V(z)^{-1}P(z))^T c = P(z)^T (V(z)^T)^{-1} c.$$

Letting

$$z = \omega,$$

and noting that the Vandermonde matrix evaluated at $\omega$ is symmetric, we have

$$(V(\omega)^T)^{-1} = V(\omega)^{-1} = \frac{1}{n}V(\omega^{-1}),$$

whose multiplication with $c$ is just the Inverse Fast Fourier transformation (IFFT) over $c$ (Geddes et al., 1992). Hence the proposition holds. ∎

## B    CALCULATION OF THEORETICAL SHAPLEY VALUES OF $R^2$

Here we will calculate the theoretical feature-specific $R^2$ values of the following three models,

$$\begin{aligned}
a. \quad & Y = \beta_1 X_1 + \beta_2 X_2 + \beta_3 X_3 + \epsilon; \\
b. \quad & Y = \beta_1 X_1 + \beta_2 X_2 + \beta_3 X_3 + \beta_4 X_1 X_2 + \beta_5 X_1 X_3 + \epsilon; \\
c. \quad & Y = \beta_1 X_1 + \beta_2 X_2 + \beta_3 X_3 + \beta_4 X_1 X_2 + \beta_5 X_1 X_2 X_3 + \epsilon.
\end{aligned}$$

With all three features generated from the Bernoulli distribution, we have

$$E(X_i) = E(X_i^2) = p_i \triangleq \mu_i,$$
$$var(X_i) = p_i(1 - p_i) \triangleq \sigma_i^2,$$
$$cov(X_1, X_1X_2) = \sigma_1^2\mu_2,$$
$$cov(X_1X_2, X_1X_3) = \sigma_1^2\mu_2\mu_3,$$
$$cov(X_1, X_1X_2X_3) = \sigma_1^2\mu_2\mu_3,$$
$$cov(X_1X_2, X_1X_2X_3) = \sigma_{12}^2\mu_3,$$
$$var(X_1X_2) = p_1p_2(1 - p_1p_2) \triangleq \sigma_{12}^2,$$
$$var(X_1X_2X_3) = p_1p_2p_3(1 - p_1p_2p_3) \triangleq \sigma_{123}^2.$$

For Model a, we have

$$var(Y) = \beta_1^2\sigma_1^2 + \beta_2^2\sigma_2^2 + \beta_3^2\sigma_3^2 + \sigma_\epsilon^2,$$
$$E(var(Y|X_1)) = \beta_2^2\sigma_2^2 + \beta_3^2\sigma_3^2 + \sigma_\epsilon^2,$$
$$E(var(Y|X_2)) = \beta_1^2\sigma_1^2 + \beta_3^2\sigma_3^2 + \sigma_\epsilon^2,$$
$$E(var(Y|X_3)) = \beta_1^2\sigma_1^2 + \beta_2^2\sigma_2^2 + \sigma_\epsilon^2,$$
$$E(var(Y|X_{\{1,2\}})) = \beta_3^2\sigma_3^2 + \sigma_\epsilon^2,$$
$$E(var(Y|X_{\{1,3\}})) = \beta_2^2\sigma_2^2 + \sigma_\epsilon^2,$$
$$E(var(Y|X_{\{2,3\}})) = \beta_1^2\sigma_1^2 + \sigma_\epsilon^2,$$
$$E(var(Y|X_{\{1,2,3\}})) = \sigma_\epsilon^2.$$

For Model b, we have

$$E(var(Y|X_\phi) = \beta_1^2\sigma_1^2 + \beta_2^2\sigma_2^2 + \beta_3^2\sigma_3^2 + \beta_4^2\sigma_{12}^2 + \beta_5^2\sigma_{13}^2 + 2\beta_1\beta_4\sigma_1^2\mu_2 + 2\beta_1\beta_5\sigma_1^2\mu_3 + 2\beta_2\beta_4\sigma_2^2\mu_1$$
$$+ 2\beta_3\beta_5\sigma_3^2\mu_1 + 2\beta_4\beta_5\sigma_1^2\mu_2\mu_3 + \sigma_\epsilon^2,$$
$$E(var(Y|X_1) = (\beta_2^2 + 2\beta_2\beta_4\mu_1 + \beta_4^2\mu_1)\sigma_2^2 + (\beta_3^2 + 2\beta_3\beta_5\mu_1 + \beta_5^2\mu_1)\sigma_3^2 + \sigma_\epsilon^2,$$
$$E(var(Y|X_2) = (\beta_1^2 + 2\beta_1\beta_4\mu_2 + \beta_4^2\mu_2)\sigma_1^2 + \beta_3^2\sigma_3^2 + \beta_5^2\sigma_{13}^2 + 2(\beta_1 + \beta_4\mu_2)\beta_5\sigma_1^2\mu_3 + 2\beta_3\beta_5\sigma_3^2\mu_1 + \sigma_\epsilon^2,$$
$$E(var(Y|X_3) = (\beta_1^2 + 2\beta_1\beta_5\mu_3 + \beta_5^2\mu_3)\sigma_1^2 + \beta_2^2\sigma_2^2 + \beta_4^2\sigma_{12}^2 + 2(\beta_1 + \beta_5\mu_3)\beta_4\sigma_1^2\mu_2 + 2\beta_2\beta_4\sigma_2^2\mu_1 + \sigma_\epsilon^2,$$
$$E(var(Y|X_{\{1,2\}}) = (\beta_3^2 + 2\beta_3\beta_5\mu_1 + \beta_5^2\mu_1)\sigma_3^2 + \sigma_\epsilon^2,$$
$$E(var(Y|X_{\{1,3\}}) = (\beta_2^2 + 2\beta_2\beta_4\mu_1 + \beta_4^2\mu_1)\sigma_2^2 + \sigma_\epsilon^2,$$
$$E(var(Y|X_{\{2,3\}}) = (\beta_1^2 + \beta_4^2\mu_2 + \beta_5^2\mu_3^2 + 2\beta_1\beta_4\mu_2 + 2\beta_1\beta_5\mu_3 + 2\beta_4\beta_5\mu_2\mu_3)\sigma_1^2 + \sigma_\epsilon^2,$$
$$E(var(Y|X_{\{1,2,3\}}) = \sigma_\epsilon^2$$

For Model c, we have

$$var(Y) = \beta_1^2\sigma_1^2 + \beta_2^2\sigma_2^2 + \beta_3^2\sigma_3^2 + \beta_4^2\sigma_{12}^2 + \beta_5^2\sigma_{123}^2 + 2\beta_1\beta_4\sigma_1^2\mu_2 + 2\beta_1\beta_5\sigma_1^2\mu_2\mu_3 +$$
$$2\beta_2\beta_4\sigma_2^2\mu_1 + 2\beta_2\beta_5\sigma_2^2\mu_1\mu_3 + 2\beta_3\beta_5\sigma_3^2\mu_1\mu_2 + 2\beta_4\beta_5\sigma_{12}^2\mu_3 + \sigma_\epsilon^2,$$
$$E(var(Y|X_1) = (\beta_2^2 + 2\beta_2\beta_4\mu_1 + \beta_4^2\mu_1)\sigma_2^2 + \beta_3^2\sigma_3^2 + \beta_5^2\mu_1\sigma_{23}^2 + 2(\beta_2 + \beta_4)\beta_5\mu_1\sigma_2^2\mu_3$$
$$+ 2\beta_3\beta_5\mu_1\sigma_3^2\mu_2 + \sigma_\epsilon^2,$$
$$E(var(Y|X_2) = (\beta_1^2 + 2\beta_1\beta_4\mu_2 + \beta_4^2\mu_2)\sigma_1^2 + \beta_3^2\sigma_3^2 + \beta_5^2\mu_2\sigma_{13}^2 + 2(\beta_1 + \beta_4)\beta_5\mu_2\sigma_1^2\mu_3$$
$$+ 2\beta_3\beta_5\mu_2\sigma_3^2\mu_1 + \sigma_\epsilon^2,$$
$$E(var(Y|X_3)) = \beta_1^2\sigma_1^2 + \beta_2^2\sigma_2^2 + (\beta_4^2 + 2\beta_4\beta_5\mu_3 + \beta_5^2\mu_3)\sigma_{12}^2 + 2\beta_1(\beta_4 + \beta_5\mu_3)\sigma_1^2\mu_2$$
$$+ 2\beta_2(\beta_4 + \beta_5\mu_3)\sigma_2^2\mu_1 + \sigma_\epsilon^2,$$
$$E(var(Y|X_{\{1,2\}}) = (\beta_3^2 + 2\beta_3\beta_5\mu_1\mu_2 + \beta_5^2\mu_1\mu_2)\sigma_3^2 + \sigma_\epsilon^2,$$
$$E(var(Y|X_{\{1,3\}}) = (\beta_2^2 + \beta_4^2\mu_1 + \beta_5^2\mu_1\mu_3 + 2\beta_2\beta_4\mu_1 + 2\beta_2\beta_5\mu_1\mu_3 + 2\beta_4\beta_5\mu_1\mu_3)\sigma_2^2 + \sigma_\epsilon^2,$$
$$E(var(Y|X_{\{2,3\}}) = (\beta_1^2 + \beta_4^2\mu_2 + \beta_5^2\mu_2\mu_3 + 2\beta_1\beta_4\mu_2 + 2\beta_1\beta_5\mu_2\mu_3 + 2\beta_4\beta_5\mu_2\mu_3)\sigma_1^2 + \sigma_\epsilon^2,$$
$$E(var(Y|X_{\{1,2,3\}}) = \sigma_\epsilon^2.$$

For all of the cases, the Shapley values can be calculated as

$$
\begin{aligned}
\phi_1 \;=\; & -\frac{1}{var(Y)}\left\{\frac{1}{3}[E(var(Y|X_1))-var(Y)]+\frac{1}{6}[E(var(Y|X_{\{1,2\}})-E(var(Y|X_2))]\right.\\
& \left.+\frac{1}{6}[E(var(Y|X_{\{1,3\}})-E(var(Y|X_3))]+\frac{1}{3}[E(var(Y|X_{\{1,2,3\}})-E(V(var|X_{\{2,3\}}))]\right\},\\
\phi_2 \;=\; & -\frac{1}{var(Y)}\left\{\frac{1}{3}[E(var(Y|X_2))-var(Y)]+\frac{1}{6}[E(var(Y|X_{\{1,2\}})-E(var(Y|X_1))]\right.\\
& \left.+\frac{1}{6}[E(var(Y|X_{\{2,3\}})-E(var(Y|X_3))]+\frac{1}{3}[E(var(Y|X_{\{1,2,3\}})-E(var(Y|X_{\{1,3\}}))]\right\},\\
\phi_3 \;=\; & -\frac{1}{var(Y)}\left\{\frac{1}{3}[E(var(Y|X_3))-var(Y)]+\frac{1}{6}[E(var(Y|X_{\{1,3\}})-E(var(Y|X_1))]\right.\\
& \left.+\frac{1}{6}[E(var(Y|X_{\{2,3\}})-E(var(Y|X_2))]+\frac{1}{3}[E(var(Y|X_{\{1,2,3\}})-E(var(Y|X_{\{1,2\}}))]\right\}.
\end{aligned}
$$

Therefore, the theoretical feature-specific $R^2$ in the three models can be evaluated and are shown in Table 1.

Table 1: Theoretical Feature-Specific $R^2$ in Simulated Models

| Model | $\sigma_\epsilon$ | Total | $X_1$ | $X_2$ | $X_3$ |
|-------|------|-------|-------|-------|-------|
|   | 0.50 | 0.9864 | 0.2094 | 0.2863 | 0.4907 |
| a | 1.00 | 0.9477 | 0.2012 | 0.2750 | 0.4715 |
|   | 1.50 | 0.8894 | 0.1888 | 0.2581 | 0.4425 |
|   | 0.50 | 0.9860 | 0.4390 | 0.1341 | 0.4129 |
| b | 1.00 | 0.9459 | 0.4212 | 0.1286 | 0.3961 |
|   | 1.50 | 0.8860 | 0.3945 | 0.1205 | 0.3710 |
|   | 0.50 | 0.9868 | 0.4288 | 0.1450 | 0.4130 |
| c | 1.00 | 0.9491 | 0.4124 | 0.1395 | 0.3972 |
|   | 1.50 | 0.8925 | 0.3878 | 0.1312 | 0.3735 |

## C    BOXPLOTS OF THE FIRST THREE FEATURE-SPECIFIC AND TOTAL $R^2$ VALUES

We have compared the performance of three different methods, i.e., our proposed Q-SHAP, SAGE by Covert et al. (2020), and SPVIM by Williamson and Feng (2020), in calculating the feature-specific $R^2$ as well as the sum of all feature-specific $R^2$ for the three models specified in Section 5, with different settings, i.e., $n \in \{500, 1000, 2000, 5000\}$, $p \in \{100, 500\}$, and $\sigma_\epsilon \in \{0.5, 1, 1.5\}$. The results are shown in Fig. 10-17. Note that the results of SAGE are unavailable in Fig. 12-17 because it cannot report those $R^2$ with our limited computational resources, and the results of SPVIM are unavailable in Fig. 14-17 because it demands too much time to complete the computation when $p = 500$.

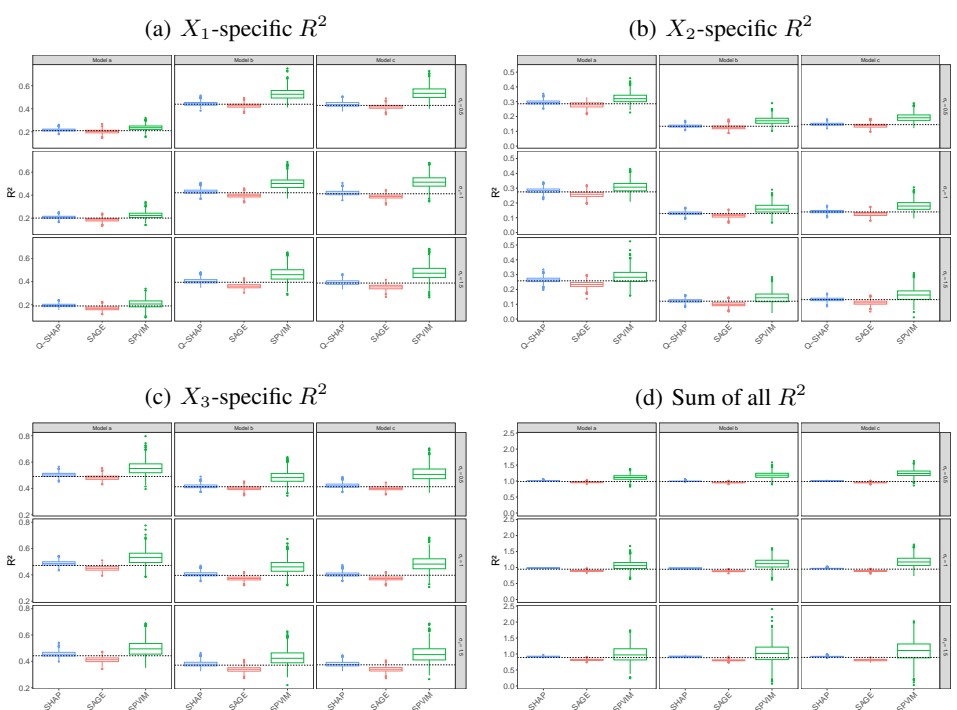

Figure 10: Boxplots of (a) $X_1$-specific, (b) $X_2$-specific, (c) $X_3$-specific, and (d) the sum of all feature-specific $R^2$ in the three models with $n = 500$, $p = 100$. The dashed lines show the theoretical $R^2$.

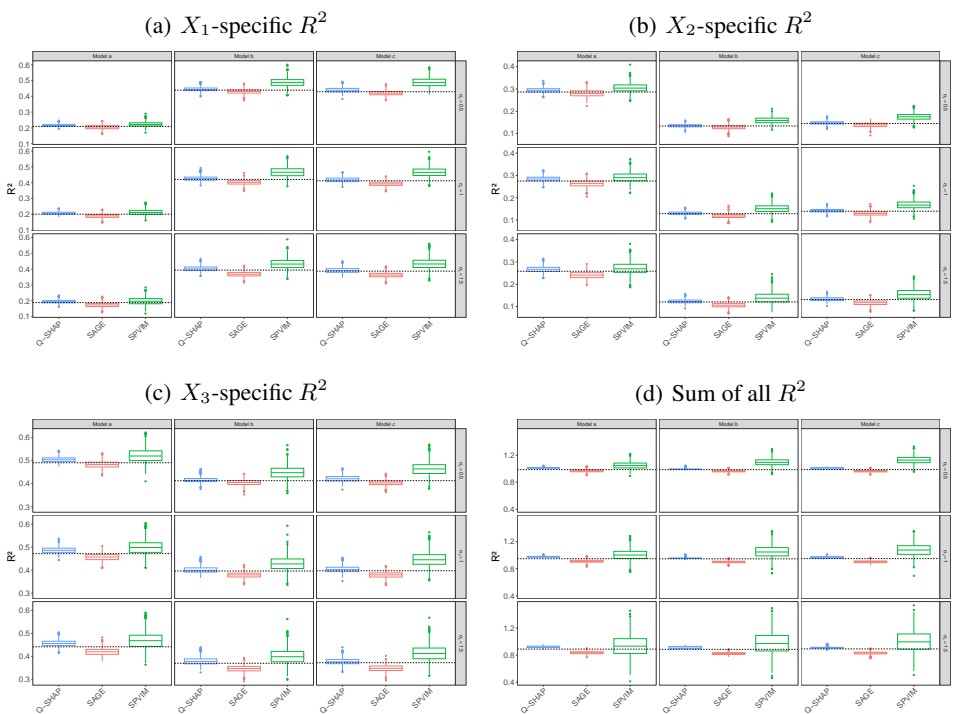

Figure 11: Boxplots of (a) $X_1$-specific, (b) $X_2$-specific, (c) $X_3$-specific, and (d) the sum of all feature-specific $R^2$ in the three models with $n = 1000$, $p = 100$. The dashed lines show the theoretical $R^2$.

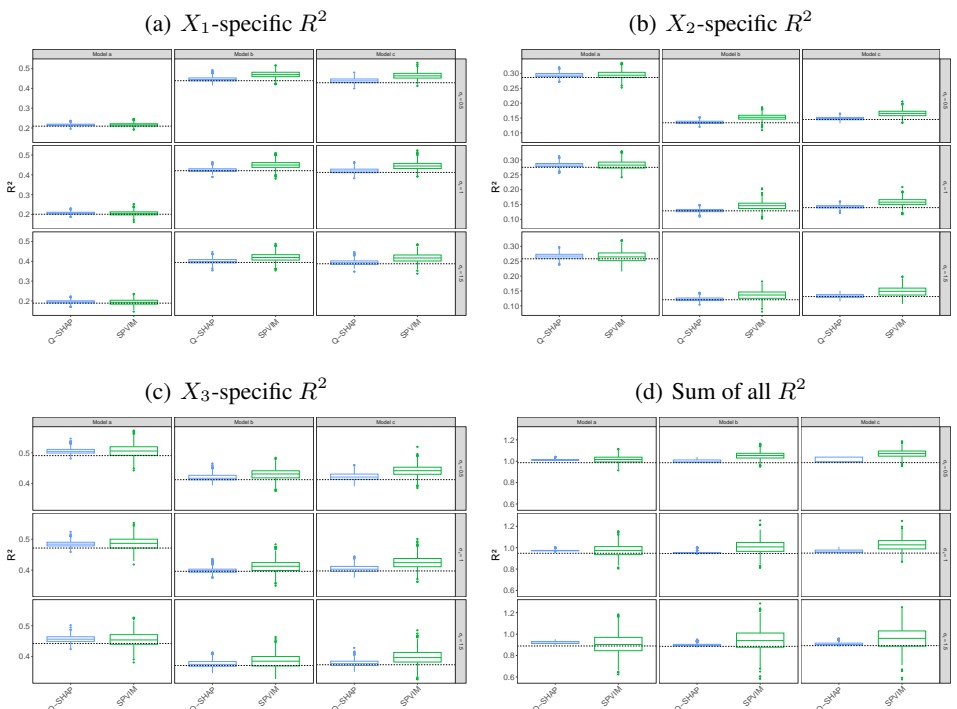

Figure 12: Boxplots of (a) $X_1$-specific, (b) $X_2$-specific, (c) $X_3$-specific, and (d) the sum of all feature-specific $R^2$ in the three models with $n = 2000$, $p = 100$. The dashed lines show the theoretical $R^2$.

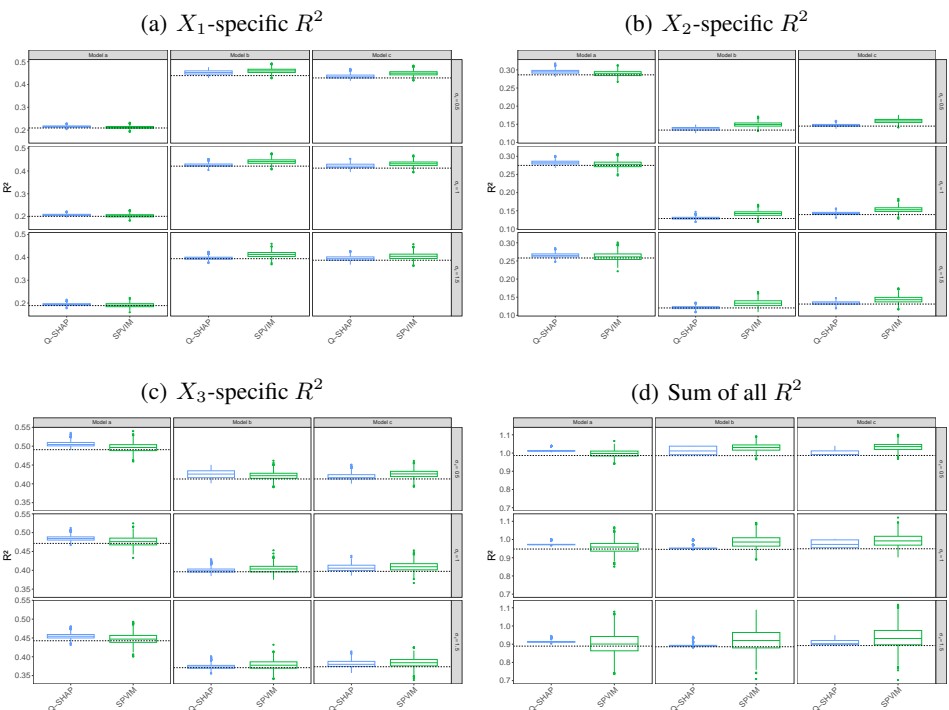

Figure 13: Boxplots of (a) $X_1$-specific, (b) $X_2$-specific, (c) $X_3$-specific, and (d) the sum of all feature-specific $R^2$ in the three models with $n = 5000$, $p = 100$. The dashed lines show the theoretical $R^2$.

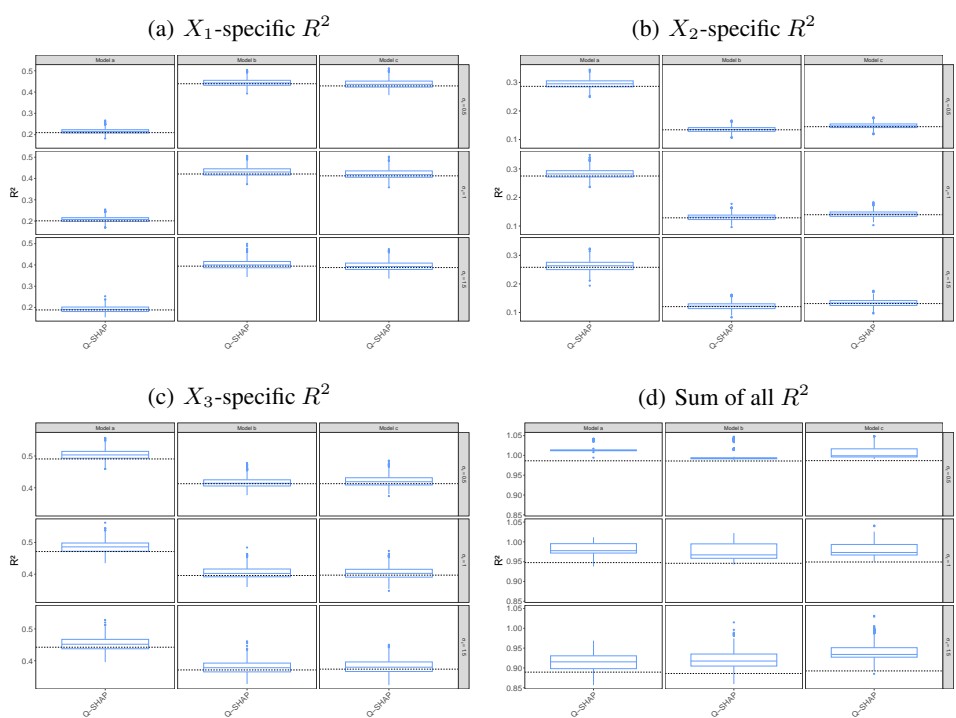

Figure 14: Boxplots of (a) $X_1$-specific, (b) $X_2$-specific, (c) $X_3$-specific, and (d) the sum of all feature-specific $R^2$ in the three models with $n = 500$, $p = 500$. The dashed lines show the theoretical $R^2$.

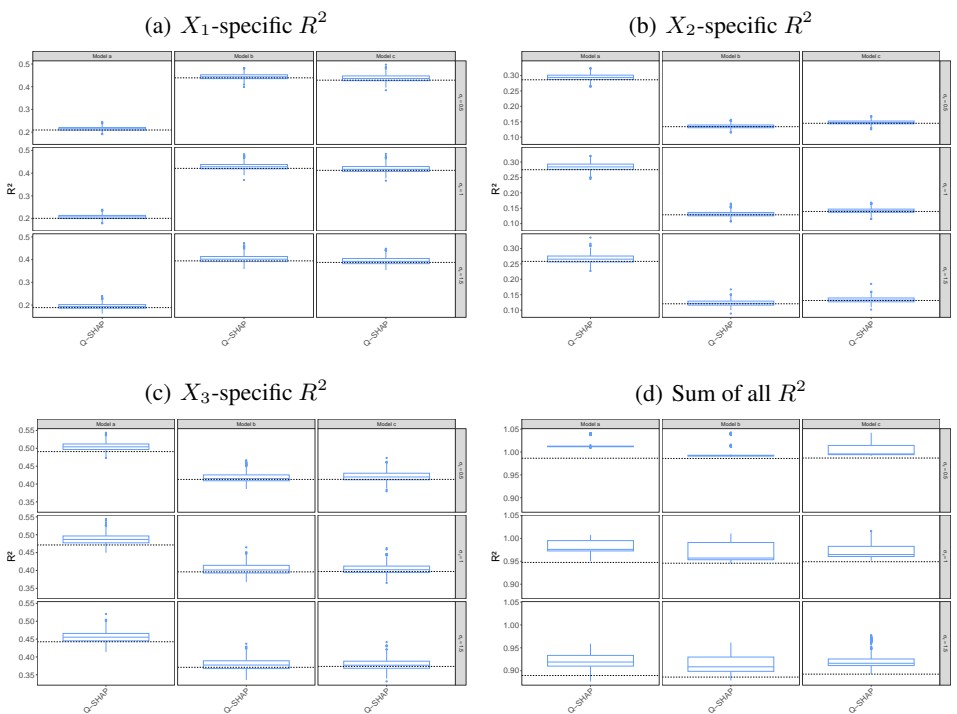

Figure 15: Boxplots of (a) $X_1$-specific, (b) $X_2$-specific, (c) $X_3$-specific, and (d) the sum of all feature-specific $R^2$ in the three models with $n = 1000$, $p = 500$. The dashed lines show the theoretical $R^2$.

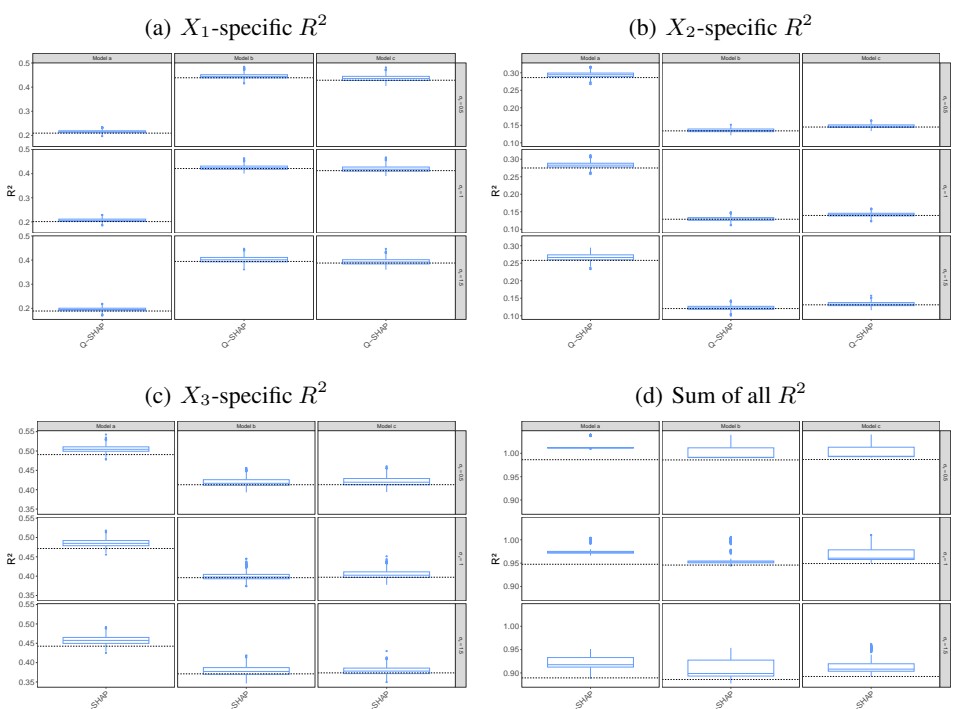

Figure 16: Boxplots of (a) $X_1$-specific, (b) $X_2$-specific, (c) $X_3$-specific, and (d) the sum of all feature-specific $R^2$ in the three models with $n = 2000$, $p = 500$. The dashed lines show the theoretical $R^2$.

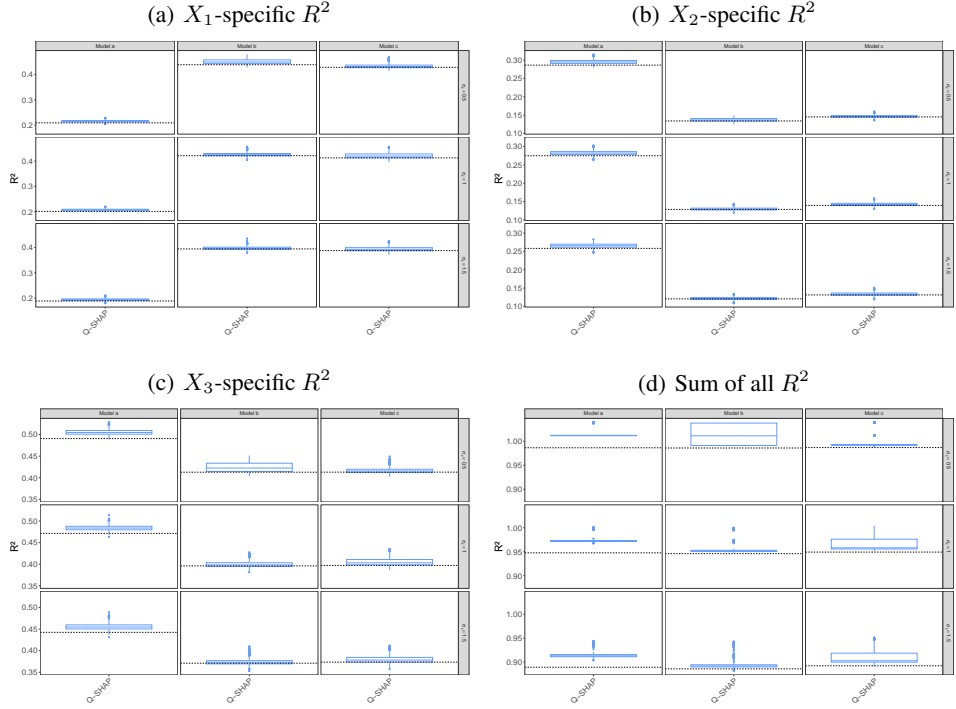

Figure 17: Boxplots of (a) $X_1$-specific, (b) $X_2$-specific, (c) $X_3$-specific, and (d) the sum of all feature-specific $R^2$ in the three models with $n = 5000$, $p = 500$. The dashed lines show the theoretical $R^2$.

## D   PLOTS OF THE MEAN ABSOLUTE ERROR (MAE)

Similar to Fig. 4, we show in Fig. 18 the mean absolute error (MAE) of feature-specific $R^2$ for both signal and nuisance features averaged over 1,000 datasets when $p = 500$. Note that the results of SAGE and SPVIM are unavailable because none of them can complete the computation for $p = 500$ with limited computational resources.

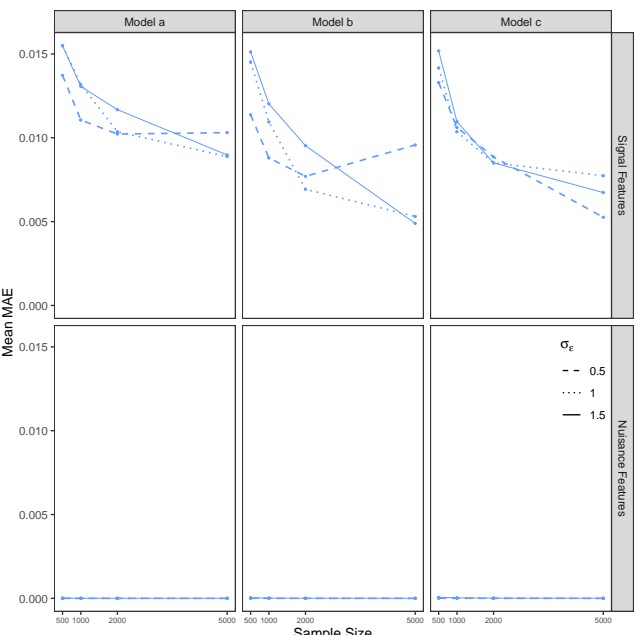

Figure 18: The mean of absolute error (MAE) of the feature-specific $R^2$ by Q-SHAP averaged across 1,000 datasets with $p = 500$

## E   COMPLEXITY OF THE ALGORITHM

Here we assume that the dataset includes $n$ samples as well as $p$ features, and a total of $T$ trees are constructed with the maximum tree depth $D$ and maximum tree leaves $L$. We denote $S$ the number of permutations taken in SAGE with $S = 10^{20}$ by default. Then, when the trees are constructed by XGBoost, the complexity of SAGE is $O(TDSpn)$ (Covert et al., 2020), and the complexity of SPVIM is $O(TDpn^2 \log n)$ (Williamson and Feng, 2020). Instead, the complexity of Q-SHAP is $O(TL^2D^2n)$ which doesn't rely on the number of features $p$.

Let's first consider the complexity of Q-SHAP in Algorithm 1 for a single tree and one sample. As shown in Figure 19, the two outer loops that iterate through the tree leaves, result in a complexity of $O(L^2)$. Within the inner loop, the computation involves the number of features induced by each pair of leaves, leading to $O(D)$ operations. The evaluation of $t[j]$ involves the computation of $C(z) \cdot P(z)$, which takes $O(D)$ operations since the number of union features between two leaves is bounded by $2D$. Combining these, the overall complexity for one tree and one sample is $O(L^2D^2)$. Thus, for the whole dataset, the complexity of Q-SHAP scales to $O(nL^2D^2)$ for a single tree. With the advancements introduced in Section 4, Q-SHAP has a total complexity of $O(TnL^2D^2)$ for the ensemble of $T$ boosting trees.

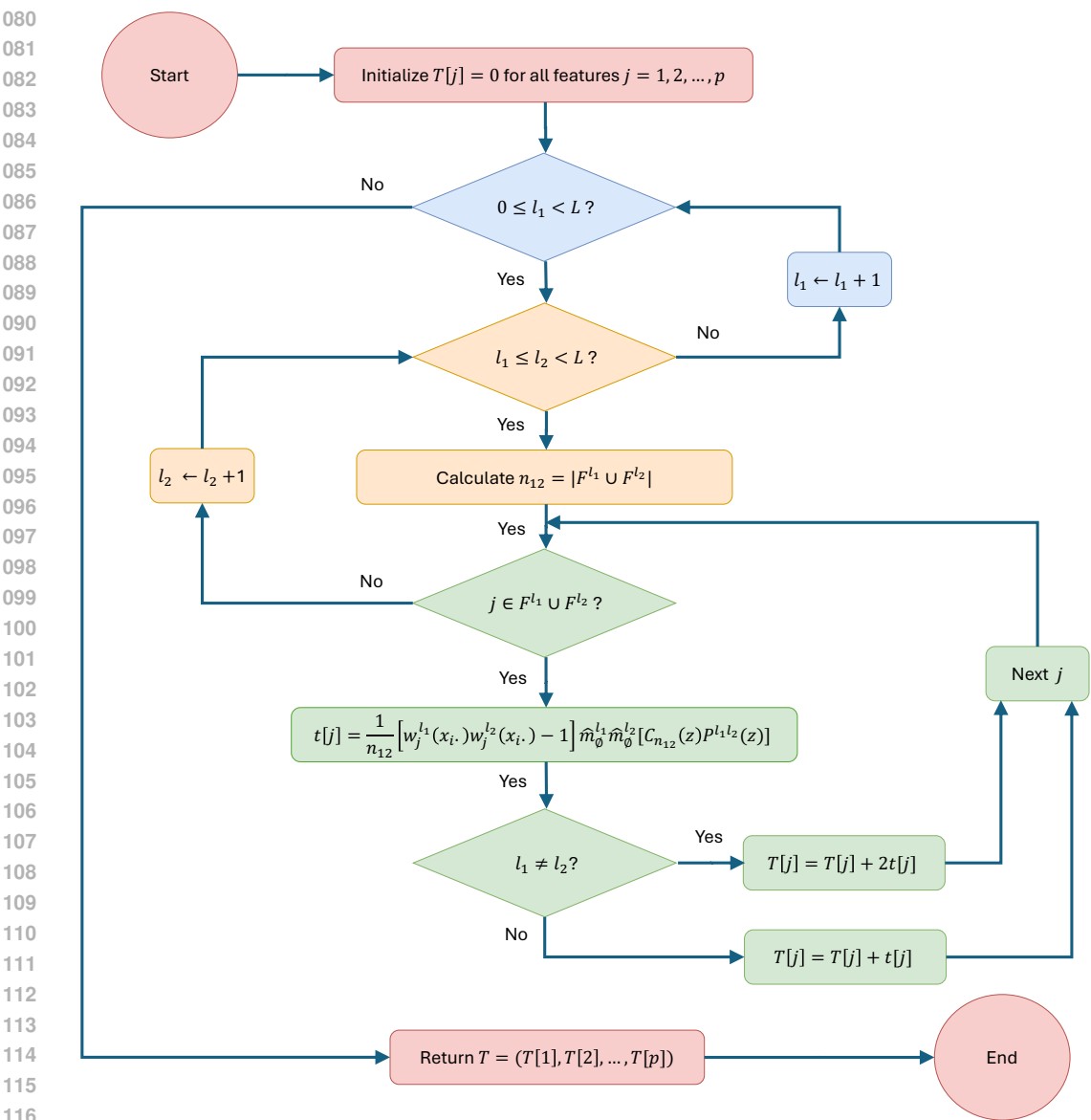

Figure 19: The Flowchart of Algorithm 1

The property that the complexity of Q-SHAP doesn't rely on the number of features is a prominent advantage of Q-SHAP and is critical in analyzing high-dimensional data. Such an advantage is achieved via Proposition 1, which eliminates dependence on $p$ by leveraging the internal structure of the tree. Furthermore, unlike SAGE and SPVIM, which require extensive sampling, Q-SHAP directly utilizes the tree's weight function introduced in Section 3.2, eliminating the need for any sampling.

## F  ANALYSES OF ADDITIONAL REAL DATA

To further illustrate the utility of our proposed method, we further apply Q-SHAP, as well as SAGE and SPVIM, to two additional publicly available datasets, i.e., healthcare insurance expenses [1] and S&P 500 stock prices from February 8, 2013 to February 7, 2018.[2] The healthcare insurance expenses

[1]https://www.kaggle.com/datasets/arunjangir245/healthcare-insurance-expenses/data
[2]https://www.kaggle.com/datasets/camnugent/sandp500/data

dataset includes eight features for each of the 1338 subjects besides their healthcare insurance costs. For the S&P 500 dataset, we focused on predicting the daily return rate of NVIDIA stock from the daily return rates of other 469 stocks which have full records of 1,258 business days.

The same settings described in Section 6 are applied to build the ensemble trees for these two datasets. To make it feasible to obtain $R^2$ from SAGE and SPVIM for the S&P 500 dataset, we applied Q-SHAP to first select the top 100 features and then rebuilt the ensemble trees with these 100 features.

As shown in Table 2, Q-SHAP is much faster than SAGE and SPVIM in calculating $R^2$ in both datasets, with SPVIM taking much longer time than the other two.

Table 2: Computing Times

|  | Q-SHAP | SAGE | SPVIM |
|---|---|---|---|
| **Heathcare Insurance Expenses** | 32 seconds | 6 minutes | 22 minutes |
| **S&P 500** | 41 seconds | 26 hours | 138 hours |

Consistent with the simulation results, SAGE tends to underestimate the feature-specific $R^2$ values and SPVIM tends to report unstable feature-specific $R^2$ values as they range from -0.0328 to 0.5646 for the healthcare insurance expenses dataset (Figure 20) and -0.3955 to 0.4571 for S&P 500 dataset (Figure 21). On the other hand, the tree ensemble for the healthcare insurance expenses data has the model $R^2$ at 0.86, and that for S&P 500 has the model $R^2$ at 0.73. However, only the sums of all feature-specific $R^2$ from Q-SHAP match these model $R^2$ values. Instead, SAGE reports 0.79 for healthcare insurance expenses and 0.50 for S&P, both are much lower than the model total $R^2$. SPVIM overestimates it for healthcare insurance expenses data with the sum of all feature-specific $R^2$ at 0.9935 but underestimates it for S&P 500 data with the sum as -0.6642.

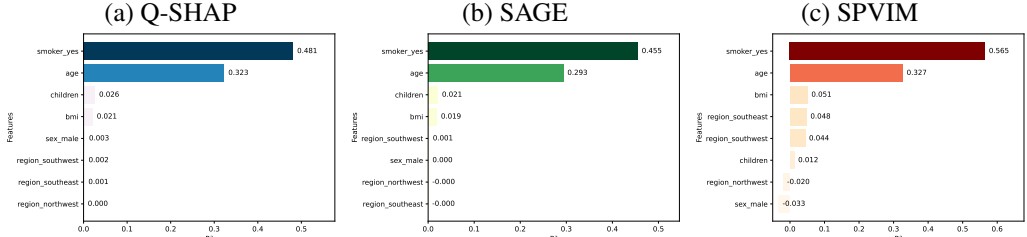

Figure 20: Feature-specific $R^2$ calculated by Q-SHAP, SAGE, and SPVIM for the healthcare insurance data.

Shown in Figure 22.(a) are the top 15 of all 469 features in the S&P 500 data with their $R^2$ calculated by Q-SHAP. For the tree ensemble built with the selected 100 features, the top 15 feature-specific $R^2$ values based on Q-SHAP, SAGE, and SPVIM are shown in 22.(b)-(d), respectively. We observed that both Q-SHAP and SAGE report the same top three features but they differ at the forth feature with Q-SHAP reporting the $R^2$ of MCHP at 0.069 but SAGE reporting the $R^2$ of MU at 0.034. On the other hand, SPVIM reports a completely different set of three features at the top and the $R^2$ of MA at 0.457. As in the case of predicting Gleason score with prostate cancer patients in Section 6, the sum of the top 15 $R^2$ by SPVIM is much larger than one.

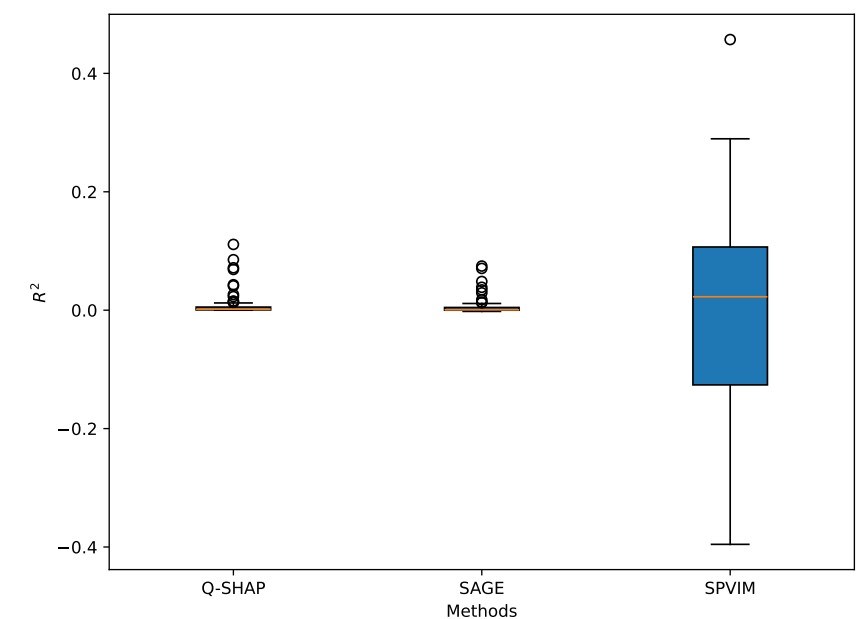

Figure 21: Feature-specific $R^2$ calculated by Q-SHAP, SAGE, and SPVIM for the S&P 500 data including only 100 features.

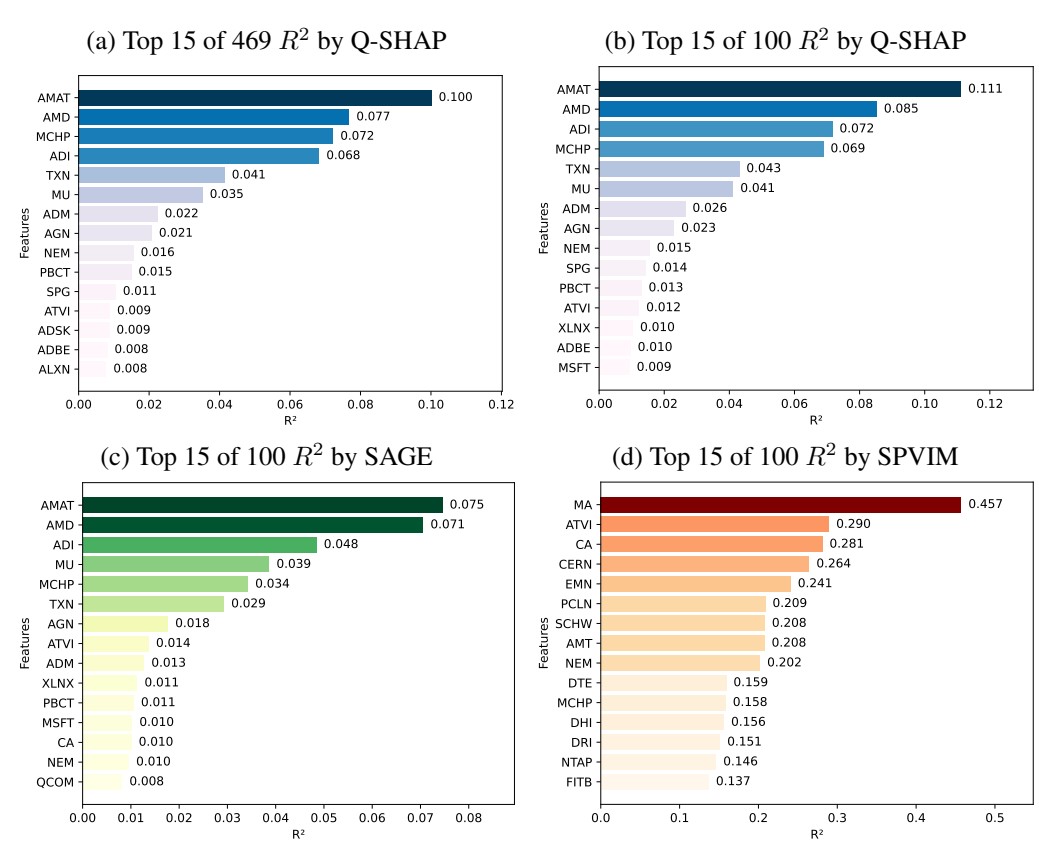

Figure 22: Top 15 features in the S&P 500 data with $R^2$ calculated by Q-SHAP, SAGE, and SPVIM

