# OpenReview forum: "Feature-Specific Coefficients of Determination in Tree Ensembles"
_ICLR.cc/2025/Conference — ICLR 2025 Conference Withdrawn Submission_

### Official Review · Reviewer_WPCk · 2024-10-30

**Soundness:** 3
**Presentation:** 4
**Contribution:** 3
**Rating:** 8
**Confidence:** 3

**Summary:**

The authors developed an algorithm Q-SHAP for fast and accurate
computation of Shapley features scores for regression problems, for
tree-based models like Gradient Boosted Machines. The algorithm is
evaluated using 3 synthetic and one real dataset, and is orders of
magnitude faster than comparator methods.

**Strengths:**

- important problem

- creative and computationally very efficient solution

- well-written manuscript

**Weaknesses:**

- relatively limited evaluation (3 synthetic datasets, one real with
  somewhat discordant results - see Questions section)

**Questions:**

For the real data (Gleason score), there is a great match between top
15 features for Q-SHAP and SAGE, but mismatch between the two and
SPVIM. Can the authors comment on this discrepancy?

---

> ### Author Response · Authors · 2024-11-23
> **Response to Reviewer WPCk**
>
> **For the real data (Gleason score), there is a great match between top 15 features for Q-SHAP and SAGE, but mismatch between the two and SPVIM. Can the authors comment on this discrepancy?**
>
> Thank you for the insightful question. Both Q-SHAP and SAGE rely on information derived from the fitted model, utilizing the predictions of pre-built trees. In contrast, SPVIM requires model refitting for sampled subset of features. In high-dimensional data settings, achieving consistent and optimal performance across models trained on different subsets of features can be challenging, even with carefully tuned hyperparameters. In general, refitting model introduces inherent variability attached to the subset of features, for example, different optimal tuning parameters may be demanded by different sets of features. This variability can lead to discrepancies in variable importance measures, where differences in performance may partly reflect inconsistencies in the refitting process rather than the true contribution of the features.

---

> > ### Comment · Reviewer_WPCk · 2024-11-25
> >
> > Thank you. No more questions/comments

---

### Official Review · Reviewer_gQXu · 2024-10-31

**Soundness:** 2
**Presentation:** 3
**Contribution:** 2
**Rating:** 3
**Confidence:** 4

**Summary:**

The paper proposes a new algorithm, Q-SHAP, designed to efficiently calculate feature-specific coefficients of determination in tree ensemble models. This algorithm addresses the computational challenges associated with Shapley values for quadratic loss, particularly in high-dimensional data contexts where calculating feature contributions becomes complex and computationally intensive. By leveraging tree structure, Q-SHAP reduces the time complexity to polynomial time, making it feasible for large datasets and models. The contributions of Q-SHAP include:

- Q-SHAP reduces complexity, enabling fast computation of feature-specific R^2.

- Extensive simulations show Q-SHAP's superiority in estimating feature-specific R^2 over existing methods.

**Strengths:**

- The paper introduces a novel algorithm, Q-SHAP, which specifically addresses the computational challenges of calculating feature-specific R^2 within tree ensembles.

- The paper explains the underlying motivation well and presents its contributions in a structured, sequential manner, making it easier for readers to follow.

- The authors conduct comprehensive simulations to test the algorithm's performance across various scenarios, demonstrating Q-SHAP’s accuracy and computational efficiency compared to well-known methods like SAGE and SPVIM.

**Weaknesses:**

- While the paper acknowledges related methods like SAGE and SPVIM, it could better contextualize Q-SHAP's theoretical advantages over these approaches. Specifically, the paper could benefit from a more detailed comparison of the mathematical foundations and computational complexity of Q-SHAP versus these methods. Including a table or diagram that compares the key theoretical aspects, limitations, and unique benefits of each method would make Q-SHAP’s improvements clearer to readers.

- The paper mentions that Q-SHAP’s framework might be extendable to other quadratic loss functions, but it doesn’t explore this idea in depth. This is a missed opportunity to underscore the broader relevance of the algorithm, as generalizability can increase impact across various applications in machine learning and data science.

- Although the paper includes a real-data example (prostate cancer gene expression data), the experiments could further explore different types of datasets to stress-test Q-SHAP’s robustness across diverse scenarios. For instance, it would be informative to see how Q-SHAP performs in datasets with imbalanced classes, highly correlated features, or extreme high-dimensional spaces.

- While the paper demonstrates that Q-SHAP is computationally faster than SAGE and SPVIM, it lacks specific breakdowns of how time complexity is reduced in different stages of the computation. Additionally, it would be useful to know how Q-SHAP’s runtime scales with an increasing number of features or trees, beyond the static benchmarks shown.

**Questions:**

The main issues have been highlighted in the section addressing weaknesses, but additional questions remain:

- Could you provide more insights or a breakdown of how polynomial time complexity is achieved within Q-SHAP? Specifically, a step-by-step breakdown of the algorithm, highlighting which specific operations contribute most significantly to reducing time complexity compared to existing methods, would clarify where the computational gains arise. This detailed explanation would make the efficiency improvements more actionable and understandable.

- Given Q-SHAP's improvements in feature-specific R^2 calculations, how does this translate to increased interpretability or trust in model predictions? Concrete examples or case studies where the improved accuracy of Q-SHAP’s feature-specific R^2 calculations enhances decision-making or model interpretation in real-world scenarios would help readers understand the practical impact of the method. For example, demonstrating applications in high-stakes fields like healthcare or finance would illustrate how Q-SHAP’s outputs can directly benefit practitioners.

- Are there limitations of Q-SHAP that practitioners should be aware of when using it on different types of tree models? A specific section dedicated to discussing Q-SHAP's limitations and potential pitfalls would be helpful. This could include guidelines on when Q-SHAP might not be the best choice, as well as precautions users should take when applying it to various tree model variants. Such a section would provide practical insights and help manage expectations for different use cases.

---

> ### Author Response · Authors · 2024-11-23
> **Response to Reviewer gQXu**
>
> **The paper mentions that Q-SHAP’s framework might be extendable to other quadratic loss functions, but it doesn’t explore this idea in depth. This is a missed opportunity to underscore the broader relevance of the algorithm, as generalizability can increase impact across various applications in machine learning and data science.**
>
> This paper mainly focuses on developing Q-SHAP for the quadratic loss, i.e., (y-\hat{y})^2, so we can calculate feature-specific R^2 for tree ensembles. We point out in “Discussion” that Q-SHAP’s framework may be extended for more general quadratic loss functions which may be of interest in practice. One example is on multivariate outcomes, such as impactful factors of a financial portfolio which consists of multiple stocks/bonds and each stock/bond may be investigated for their important features via tree ensembles. The quadratic loss of the portfolio may be considered as (\sum_k w_k y_k-\sum_k w_k \hat{y}_k)^2 with w_k and y_k the weight and price/return for k-th stock, which applies to the general case of multivariate outcomes.
>
> **For instance, it would be informative to see how Q-SHAP performs in datasets with imbalanced classes, highly correlated features, or extreme high-dimensional spaces.**
>
> For datasets with imbalanced classes, we would like to point out that classical definition of R^2 is known to be bounded at 0.75 for binary outcomes and thus R^2 may be modified to be non-quadratic functions, see Cameron &Windmeijer (1997) and Zhang (2017). We will extend our work for such measures in a separate project.
>
> Cameron, A. Colin, and Frank AG Windmeijer. "An R-squared measure of goodness of fit for some common nonlinear regression models." Journal of Econometrics 77.2 (1997): 329-342.
>
> Zhang, Dabao. "A coefficient of determination for generalized linear models." The American Statistician 71.4 (2017): 310-316.
>
> **Given Q-SHAP's improvements in feature-specific R^2 calculations, how does this translate to increased interpretability or trust in model predictions? Concrete examples or case studies where the improved accuracy of Q-SHAP’s feature-specific R^2 calculations enhances decision-making or model interpretation in real-world scenarios would help readers understand the practical impact of the method. For example, demonstrating applications in high-stakes fields like healthcare or finance would illustrate how Q-SHAP’s outputs can directly benefit practitioners.**
>
> As feature-specific R^2 intends to measure the proportion of total variation in the outcome explained by a single feature, Q-SHAP facilitates such calculation by simplification via theoretical results shown in Propositions 1 & 2 and Theorem 1. We have added two more datasets to showcase the utility of our proposed methods, one on the Healthcare Insurance Expenses dataset and another on S&P 500 Stock Prices dataset.
>
> **Are there limitations of Q-SHAP that practitioners should be aware of when using it on different types of tree models? A specific section dedicated to discussing Q-SHAP's limitations and potential pitfalls would be helpful. This could include guidelines on when Q-SHAP might not be the best choice, as well as precautions users should take when applying it to various tree model variants. Such a section would provide practical insights and help manage expectations for different use cases.**
>
> Q-SHAP is currently designed to work with any trees and ensembles of boosted trees, leveraging their structure and properties to compute exact quadratic SHAP values. Practitioners should be aware that R^2 is a well-defined metric for regression tasks, ranging between 0 and 1. However, for binary classification tasks, classical R^2 is bounded by 0.75. Many works have been proposed to address such boundedness as shown in Cameron &Windmeijer (1997) and Zhang (2017).
>
> Cameron, A. Colin, and Frank AG Windmeijer. "An R-squared measure of goodness of fit for some common nonlinear regression models." Journal of Econometrics 77.2 (1997): 329-342.
>
> Zhang, Dabao. "A coefficient of determination for generalized linear models." The American Statistician 71.4 (2017): 310-316.

---

> > ### Comment · Reviewer_gQXu · 2024-11-26
> >
> > I sincerely thank the authors for their detailed responses. After carefully reviewing them, I find that the responses do not fully address my concerns. Additionally, the authors have used a linear dataset in the simulation section, while real-world datasets are more likely to exhibit non-linear characteristics. This makes the use of R^2 an inappropriate metric in such settings. Therefore, I recommend that the authors revise the paper to address these issues and consider resubmitting it.

---

> ### Author Response · Authors · 2024-11-26
> **Second Response to Reviewer gQXu**
>
> **I sincerely thank the authors for their detailed responses. After carefully reviewing them, I find that the responses do not fully address my concerns. Additionally, the authors have used a linear dataset in the simulation section, while real-world datasets are more likely to exhibit non-linear characteristics. This makes the use of R^2 an inappropriate metric in such settings. Therefore, I recommend that the authors revise the paper to address these issues and consider resubmitting it.**
>
>
> We have been through every point of all reviewers and carefully addressed in our responses. Please be aware that, for three common concerns, we have addressed in the top “Response to Common Concerns”, which includes “Synthetic Data Analyses with Ground Truth”.
>
> We considered three different models, all with binary features for the purpose of easily mapping the original models into tree models and calculating theoretical feature-specific R^2 values as shown in Table 1. For comparison, we start with a simple linear regression in the model a, however, the models b & c are not linear in terms of the interested features. We consider these three models sufficiently serve our purpose of demonstrating the advantages and disadvantages of different methods. However, it’s not our purpose here to cover all related issues, which could be too ambitious.
>
> We don’t understand why R^2 is an inappropriate metric for nonlinear models (please let us know if we misunderstand your point). One of the promising advantages of Shapley values is their efficiency even for nonlinear models, which is the reason we calculate the feature-specific R^2 in the sense of Shapley values.

---

### Official Review · Reviewer_cDvG · 2024-11-02

**Soundness:** 3
**Presentation:** 3
**Contribution:** 3
**Rating:** 6
**Confidence:** 4

**Summary:**

The authors present Q-SHAP, a polynomial-time algorithm for computing Shapley-valued-based individual feature contributions to R^2 for tree ensemble models for a dataset in aggregate (as opposed to using Shapley to explain individual predictions). R^2 decomposition is expressed in terms of Shapley values for model predictions and squared model predictions for individual data points and then summed over the data. The authors leverage the piecewise-constant prediction surface of trees within individual leaves and apply the inverse fast fourier transform to achieve polynomial-time computation .Q-SHAP is shown to do a better job than 2 competing methods of estimating R^2 feature importance for a collection of toy problems of 3 variables where ground truth is known.  Q-SHAP is also illustrated on one real world bioinformatics gene expression problem where the number of features (17k) far exceeds the number of data points (551).

**Strengths:**

The work is original to the best of my knowledge. Given the large volume of ML papers in recent years, it's possible that there are other polynomial time methods for the same problem, but I'm not aware of them.

I think the paper clearly has enough significance to merit acceptance at ICLR. Tree ensembles such as gradient boosting machines (XGBoost, etc) continue to be state of the art for tabular problems, as evidenced in their prevalence in winning Kaggle competitions. Feature importance is certainly a significant topic for such models. It's quite worthwhile to be able to do a Shapley decomposition at the dataset level to explain R^2 contributions of features for such models.

The clarity of the paper could be improved (see my comments in weaknesses) but is near the adequate range for acceptance.

**Weaknesses:**

I think the experimental results are a little thin. There is only one real-world example in the paper. I realize that you can't really get ground-truth for feature importance for real world problems, so that does make it somewhat less important to have results on a large collection of real datasets than for a paper where predictive accuracy is the main contribution. Even so, I would have preferred to see the paper illustrated on at least one more problem. The gene expression problem seems quite extreme in terms of having number of features >>> number of data points. I realize that large feature sets may be the area where Q-SHAP shines, but results on a 2nd problem which is not such an outlier in terms of number of features would have been nice.

While I realize that there is plenty of interest in Shapley and that Shapley is the unique solution to a set of key desiderata, I also would have appreciated more illustration of the failings of a method like feature permutation importance, given how prominent that method is. I'm not familiar with the Ishwaran paper that is cited, but it's not in a well known ML journal (by the way Electronic Journal of Statistics has a misspelling in the citations.."Elecctronic" ).

I also have several typo/rewrite suggestions:

page 2

for any set of feature -> for any set of features

and dummy? "null player" is I think a more standard term for that Shapley criterion.

page 3

lies on the fact -> lies in the fact

Fig 2 caption hypothetic -> hypothetical

Page 8

maximum depth of Models -> maximum depth of models

Page 9
based the result-> based on the result

quadartic -> quadratic

**Questions:**

Do you have results on another real-world problem?

Can you explain in more detail the failings of permutation importance and how a Shapley approach would be better, with a concrete example?

---

> ### Author Response · Authors · 2024-11-23
> **Response to Reviewer cDvG**
>
> **While I realize that there is plenty of interest in Shapley and that Shapley is the unique solution to a set of key desiderata, I also would have appreciated more illustration of the failings of a method like feature permutation importance, given how prominent that method is. I'm not familiar with the Ishwaran paper that is cited, but it's not in a well known ML journal (by the way Electronic Journal of Statistics has a misspelling in the citations.."Elecctronic" ).**
>
> Thanks for pointing out this typo as well as other typos.
>
> We agree that the feature permutation importance (PFI) is an interesting concept. PFI is a model-agnostic technique and it measures the increase in the model's prediction error after permuting the values of a feature, with increase size indicating the model’s reliance on that feature. Electronic Journal of Statistics is a decent journal in statistics. Rather than the article by Ishwaran (2007), we would refer to some disadvantages of PFI mentioned in Chapter 8.5.5 by Molnar (2020) with examples.
>
> Unlike R^2, which ranges from 0 and 1 and can be interpreted as a percentage of explained variance, PFI lacks a standardized scale, making it unclear how to interpret the importance of a feature in the context of a specific model or determine statistically meaningful cutoffs.
>
> **I also have several typo/rewrite suggestions: … dummy? "null player" is I think a more standard term for that Shapley criterion.**
>
> Thanks for pointing out those typos which we have corrected in the updated manuscript.
>
> For the Shapley value, Owen and Prieur (2017) explicitly stated its “dummy” property and we would like to keep our use of “dummy” for consistency.
>
> **Can you explain in more detail the failings of permutation importance and how a Shapley approach would be better, with a concrete example?**
>
> We calculated all measures of all eight features in the Health Insurance Expenses dataset, and presented the results in the following, with SW for region:southwest, SE for region:southeast, NW for region: northwest, and PFI for permutation feature importance.
>
>       Method     smoker     age      children     BMI       sex         SW        SE        NW
>       Q-SHAP     0.4807    0.3226      0.0260    0.0208    0.0029    0.0019    0.0014    0.0000
>       SAGE       0.4546    0.2932      0.0210    0.0187    0.0002    0.0012   -0.0003    0.0000
>       SPVIM      0.5646    0.3266      0.0124    0.0507   -0.0328    0.0439    0.0479   -0.0199
>       PFI      1159.0293  739.0218    71.6984   64.0318    5.3566    5.1032    3.0180    0.1565
>
> Interestingly, the ranking produced by PFI aligns perfectly with that of Q-SHAP. Let’s take smoker to illustrate the advantage of R^2 over PFI. Q-SHAP reports smoker-specific R^2 at 0.4807, implying “smoker” itself accounts for more than 48% of the variation in observed insurance expenses. Although permuting smoker resulted in an increase of squared prediction errors with 1159.0293, which is difficult to interpret. On the other hand, we may target a model to achieve 80% explained variation so to include both smoker and age based on R^2 calculated by Q-SHAP while PFI may have to be targeted for a user-chosen cutoff or top features.

---

> > ### Comment · Reviewer_cDvG · 2024-11-24
> > **I will keep my score the same**
> >
> > Thanks for the responses. I will keep my score the same. I appreciate the additional experiment. However, I'm not sure the non-normalized output of PFI is a meaningful drawback given that you can easily normalize it by the variance of the target.

---

> > > ### Author Response · Authors · 2024-11-26
> > > **Second Response to Reviewer cDvG**
> > >
> > > **Thanks for the responses. I will keep my score the same. I appreciate the additional experiment. However, I'm not sure the non-normalized output of PFI is a meaningful drawback given that you can easily normalize it by the variance of the target.**
> > >
> > > Thanks for the interesting point. As it is unknown what a PFI measures, simply normalizing it may still have the same issue. In fact, we also calculated the PFI based on R^2 (previously we calculated SSE). As shown in the following of results for the healthcare insurance expenses data, the PFI-R^2 of smoker exceeds 1, which is much larger than the R^2 values reported by other methods. In particular, we don’t have a good interpretation of these changes of R^2 values after permutating variables.
> > >
> > >       Method     smoker     age      children     BMI       sex         SW        SE        NW
> > >       Q-SHAP     0.4807    0.3226      0.0260    0.0208    0.0029    0.0019    0.0014    0.0000
> > >       SAGE       0.4546    0.2932      0.0210    0.0187    0.0002    0.0012   -0.0003    0.0000
> > >       SPVIM      0.5646    0.3266      0.0124    0.0507   -0.0328    0.0439    0.0479   -0.0199
> > >       PFI-SSE  1159.0293  739.0218    71.6984   64.0318    5.3566    5.1032    3.0180    0.1565
> > >       PFI-R^2    1.0253    0.6537      0.0634    0.0566    0.0047    0.0045    0.0027    0.0001

---

### Official Review · Reviewer_nZh9 · 2024-11-04

**Soundness:** 2
**Presentation:** 2
**Contribution:** 2
**Rating:** 3
**Confidence:** 3

**Summary:**

The paper proposes an extension of Shapley values to decision tree ensembles for quadratic losses for the whole training set. Similar to TreeSHAP, it exploits the tree hierarchy in calculation. Experiments are performed on simulations studies and on a real genome data.

**Strengths:**

The is the first paper that adapts Shapley values for tree ensembles for quadratic losses, which can be used to estimate coefficient of determination globally for the whole training set.

**Weaknesses:**

1. The motivation of the paper is not clear. Shapley values by itself are quite controversial in terms of an interpretability tool, and using it globally to estimate coefficient of determination is not clear.
2. The paper is hard to follow. The actual algorithm is presented in Sections 3.3 and Section 4 which is only one page. It is not obvious how it is different from the previous work.

**Questions:**

Does the algorithm exactly estimate the Shapley values or approximately?

---

> ### Author Response · Authors · 2024-11-23
> **Response to Reviewer nZh9**
>
> **The motivation of the paper is not clear. Shapley values by itself are quite controversial in terms of an interpretability tool, and using it globally to estimate coefficient of determination is not clear.**
>
> Please find the statement of our motivation for our research in Lines 41-61. We agree that the Shapley value has some limitations. However, they remain highly useful in practice, due to its foundational properties of fairness and consistency. With the coefficient of determination a widely used metric for assessing the goodness of fit of a model with the given set of features, its decomposition to account for individual features’ contributions has been rarely explored. We hope that our work to employ fairness properties of Shapley values and define feature-specific coefficient of determination addresses practical issues in model explainability.
>
> **The paper is hard to follow. The actual algorithm is presented in Sections 3.3 and Section 4 which is only one page. It is not obvious how it is different from the previous work.**
>
> We agree that some parts of the paper are very technical. We therefore developed whole Section 3 to explain our development of Q-SHAP for single trees, and Section 4 for tree ensembles. With the technical development detailed in Section 3.2, the algorithm itself can be neatly and concisely described in Section 3.3 for single trees.
>
> In Lines 46-48, we stated that existing works, i.e., Covert et al. on SAGE and Williamson & Feng (2020) on SPVIM, both are Monte Carlo-based. We further stated issues with these Monte Carlo-based methods in Lines 49-51, which prompted our development of model-based method Q-SHAP to leverage the model structures. In summary, our algorithm takes advantage of theoretical results such as Propositions 1 & 2 and Theorem 1 to simplify the complicated calculation with our algorithm Q-SHAP having polynomial complexity.
>
> **Does the algorithm exactly estimate the Shapley values or approximately?**
>
> Our algorithm Q-SHAP takes advantage of the tree structure and provides exact estimates of the Shapley values by taking advantage of theoretical results to achieve the polynomial complexity.

---

### Official Review · Reviewer_kwC8 · 2024-11-04

**Soundness:** 3
**Presentation:** 3
**Contribution:** 3
**Rating:** 8
**Confidence:** 3

**Summary:**

Coefficients of determination, R2,  represents the total variation explained by a feature. However, its computation for the datasets with huge number of features is challenging due to the underlying quadratic loss. This study introduces the Q-SHAP algorithm, a novel approach for calculating Shapley values in tree ensembles to decompose R2 and  speed up its computation.  The goal is to reduce the computational complexity to polynomial time when calculating Shapley values related to quadratic loss terms. The proposed approach leverages tree structure, polynomial-based calculations and the Inverse Fast Fourier Transform (IFFT).

Polynomial representation is computed by defining a polynomial based on the product of weights for features along the paths to leaves. Specifically, for each pair of leaves, calculate the union of their feature sets and the size of this union to identify the features involved in the paths to these leaves. To calculate the contribution for each feature in the combined feature sets, the algorithm considers a normalization coefficient to balance the contribution of each feature, proportional to the total number of features in the combined set. After applying a weight adjustment, the algorithm predicts each feature’s individual impact on the overall prediction by estimating the contributions through the coefficient polynomial, which aggregates these contributions to simplify the prediction process for the subset of features.
The polynomial representation of feature contributions allows QSHAP to take advantage of IFFT, which is helpful for calculating the dot product of the polynomial representations of feature contributions. By applying IFFT, it  can quickly compute aggregated Shapley values over multiple leaves and trees, which is faster than directly summing or iterating over each possible combination. The paper has conducted experimentation on synthetic as well as real datasets.

**Strengths:**

Scalability of the proposed approach: QSHAP achieves polynomial-time complexity, making it computationally feasible to use in large datasets and high-dimensional models. This efficiency is particularly advantageous compared to previous Shapley value methods that are NP-hard in general.

Theoretical foundation: The authors provide mathematical proofs, such as the use of polynomial forms.

**Weaknesses:**

Dependency on Tree Structure: The algorithm’s design leverages specific tree-based model properties, consequently, QSHAP may not directly generalize to non-tree ensemble methods.

The authors need to simplify and better explain the part that they have computed the polynomial representation, using meaningful step wise flowchart, figures, etc.

**Questions:**

The part that you explain about predicting polynomial representation needs a one paragraph summary in the beginning with a justification of the choices you have made, all in one place. The reader gets some ideas before going through all the formulas. Specially explaining how do you handle situations where features appear multiple times along the paths to the leaves? Does this affect the weight assignment for these features?

---

> ### Author Response · Authors · 2024-11-23
> **Response to Reviewer kwC8**
>
> **Dependency on Tree Structure: The algorithm’s design leverages specific tree-based model properties, consequently, QSHAP may not directly generalize to non-tree ensemble methods.**
>
>  We intend to work out a way for fast calculation of quadratic loss, particularly R^2, for tree ensembles as, in practice, importance evaluation of single or a set of features are troubled with choice of cutoff. R^2 instead provides a direct evaluation for the features’ explainability of total variation in the outcome, and users can easily set up their preferred percentage value as a cutoff.
>
>  **The part that you explain about predicting polynomial representation needs a one paragraph summary in the beginning with a justification of the choices you have made, all in one place. The reader gets some ideas before going through all the formulas. Specially explaining how do you handle situations where features appear multiple times along the paths to the leaves? Does this affect the weight assignment for these features?**
>
> Thanks for the suggestion. We have explained the overall idea in Lines 148-152 and also taken Figures 1 & 2 to illustrate the main idea. We agree that the flow from Line 208 until the end of Section 3.2 is rather technical. We have tried our best to make it mathematically clear while concise to fit the page limit. We welcome any insight and suggestions for better presentation of this part.
>
> The formula in Lines 210-214 provides the calculation of weights, which can be obtained by recursive application of the idea illustrated in Lines 191 –203. In particular, the calculation in Lines 210 – 211 involves c for c-th appearance of the j-th feature and the weight is the product of sample size ratios across different appearances. The notations involved in this calculation can be found in Lines 204-207. In summary, the multiple appearances of features in different paths to the leaves DO affect the weight assignment for these features.

---

> > ### Comment · Reviewer_kwC8 · 2024-11-25
> > **Thank you for your comments**
> >
> > Thank you for your comments. I keep my score, as also the draft looks pretty much the same as the first version.

---

> > > ### Author Response · Authors · 2024-11-26
> > > **Second Response to Reviewer kwC8**
> > >
> > > **Thank you for your comments. I keep my score, as also the draft looks pretty much the same as the first version.**
> > >
> > > We appreciate your comments. As we have stated in the “Response to Common Concerns”, we have added Appendix E to detail the calculation of algorithmic complexity of our proposed algorithm and comment on its advantage over SAGE and SPVIM. We also added a flowchart for Algorithm 1 for better visualization.
> > >
> > >  We also included analyses of two additional real data in Appendix F, one with limited number of features (i.e., only 8 features), and another one with 469 features. Because of limited running time, we trimmed the 469 features to 100 features so we could apply both SAGE and SPVIM for comparison.
> > >
> > >
> > >
> > > For the main text, we also corrected some typos.

---

### Comment · Area_Chair_9AAH · 2024-11-13
**authors - reviewers discussion open until November 26 at 11:59pm AoE**

Dear authors & reviewers,

The reviews for the paper should be now visible to both authors and reviewers. The discussion is open until November 26 at 11:59pm AoE.

Your AC

---

> ### Comment · Area_Chair_9AAH · 2024-11-25
>
> Dear reviewers,
>
> The authors have provided individual responses to your reviews. Can you acknowledge you have read them, and comment on them as necessary? The discussion will come to a close very soon now:
> - Nov 26: Last day for reviewers to ask questions to authors.
> - Nov 27: Last day for authors to respond to reviewers.
>
> Your AC

---

### Author Response · Authors · 2024-11-23
**Response to common concerns**

Dear Reviewers,

We sincerely thank you for your thoughtful and detailed feedback on our manuscript. Your comments have given us an excellent opportunity to address concerns and refine the quality of our work. We have carefully reviewed each point raised and made revisions accordingly to improve the clarity and rigor of the manuscript.

We would like to address three common concerns here and address the left ones in response to corresponding reviewers. We would like to let you know that we have implemented Q-SHAP in Python and are ready to publish it via GitHub once the current review is complete.

1) Polynomial Complexity of Q-SHAP and Advantages over SAGE and SPVIM:

Here’s our complexity calculation of Q-SHAP (Algorithm 1) for a single tree and one sample: The two outer loops that iterate through the L tree leaves, resulting in a complexity of O(L^2). Within the inner loop, the computation involves the number of features induced by each pair of leaves, which is O(D) .The evaluation of  t[j]  involves the computation of  C(z) \cdot P(z) , which takes O(D) operations since the number of union features between two leaves is bounded by 2D. Combining these, the overall complexity for one tree and one sample is  O(L^2D^2) . For a dataset with sample size n, the complexity of Q-SHAP scales to O(nL^2D^2) for a single tree. With the advancements introduced in Section 4, Q-SHAP has a total complexity of O(TnL^2D^2) for an ensemble of T boosting trees.

For comparison, we list the computation complexities of all three methods in the below:

Q-SHAP:   O(T*L^2*D^2*n)

SAGE:       O(T*D*S*p*n)

SPVIM:     O(T*D*p* n^2*logn)

where

n — sample size;

T — number of trees;

D — maximum tree depth;

L — maximum number of tree leaves;

p — Number of features;

S — Number of permutations (S=10^20 in SAGE by default).


A prominent advantage of Q-SHAP is that the computation does not depend on the number of features p, which is critical in analyzing high-dimensional data. Such an advantage is achieved via Proposition 1, which eliminates dependence on p by leveraging the internal structure of the tree. Furthermore, unlike SAGE and SPVIM, which require extensive sampling, Q-SHAP directly utilizes the tree’s weight function introduced in Lines 210-214, eliminating the need for any sampling.

We have added Appendix E to provide such details.

2) Synthetic Data Analyses with Ground Truth:

Because of the concern on the ground-truth, we carefully designed three true models for the simulation study so that, for each model, we can obtain theoretical values of all feature-specific R^2 values. To mimic practical challenges, we have added different numbers of noisy features (resulting in a total of 100 and 500 features) as well as different sample sizes (500, 1000, 2000, 5000) and different levels of noise (with error standard deviation at 0,5, 1, 1.5). For each of these 72 different configurations, we simulated 1,000 datasets, leading to a total of 72,000 datasets. We have summarized the results in Section 5 of the main text and Section C of Appendix.

---

> ### Author Response · Authors · 2024-11-23
> **Response to common concerns (continued)**
>
> 3) Additional Analyses of Two Publicly Available Datasets:
>
> i) Healthcare Insurance Expenses (HIE) available at https://www.kaggle.com/datasets/arunjangir245/healthcare-insurance-expenses/data;
>
> ii) S&P500 Stock Prices (S&P) available at  https://www.kaggle.com/datasets/camnugent/sandp500/data.
>
> The HIE dataset includes eight features for each of 1338 subjects besides their healthcare insurance costs. For the S&P dataset, we focused on predicting the daily return rate of NVIDIA stock from the daily return rates of other 469 stocks which have full records of 1,258 business days from February 8, 2013 to February 7, 2018.
>
> The same settings described in “Real Data Analysis” are applied to build the ensemble trees for these two datasets. To make it feasible to obtain R^2 from SAGE and SPVIM for the S&P500 dataset, we applied Q-SHAP to first select the top 100 features and then rebuilt the ensemble trees with these 100 features. We report the computing times in the following, showing that Q-SHAP is much faster than the other two:
>
>                         Computing Time
>                 Q-SHAP             SAGE           SPVIM
>       HIE     32 seconds        6 minutes      22 minutes
>       S&P     41 seconds        26 hours       138 hours
>
>
> We also report the summary statistics of feature-specific R^2 calculated by each method in the following:
>
>                                Summary Statistics of Feature-Specific R^2
>                          Min         Q1      Median        Q3        Max          Sum
>       HIE:   Q-SHAP     0.0000     0.0018    0.0118      0.1002    0.4807       0.8565
>              SAGE      -0.0003     0.0002    0.0099      0.0890    0.4546       0.7884
>              SPVIM     -0.0328     0.0043    0.0459      0.1196    0.5646       0.9935
>       S&P:   Q-SHAP     0.0000     0.0005    0.0016      0.0053    0.1111       0.7333
>              SAGE      -0.0020     0.0000    0.0006      0.0046    0.0746       0.4993
>              SPVIM     -0.3955    -0.1262    0.0227      0.1068    0.4571      -0.6642
>
> Consistent with the simulation results, SAGE tends to underestimate the feature-specific R^2 values and SPVIM tends to report unstable feature-specific R^2 values as they range from -0.0328 to 0.5646 for HIE dataset and -0.3955 to 0.4571 for S&P dataset. On the other hand, the tree ensemble for HIE has the model R^2 at 0.86, and that for S&P has the model R^2 at 0.73. However, only the sums of all feature-specific R^2 from Q-SHAP match these model R^2 values as shown in the above table. Instead, SAGE reports 0.79 for HIE and 0.50 for S&P, both are much lower than the model total R^2. SPVIM instead overestimates it for HIE with the sum of all feature-specific R^2 at 0.9935 but underestimates it for S&P with the sum to -0.6642.
>
> We have added Appendix F to include the results on these two datasets.

---

### Note · Authors · 2025-01-28

I have read and agree with the venue's withdrawal policy on behalf of myself and my co-authors.